# BRIDGE-TTS: TEXT-TO-SPEECH SYNTHESIS WITH SCHRODINGER BRIDGE

## ABSTRACT

In text-to-speech (TTS) synthesis, diffusion models have achieved promising generation quality. However, because of the pre-defined data-to-noise diffusion process, their prior distribution is restricted to a noisy representation, which provides little information of the generation target. In this work, we present a novel TTS system, Bridge-TTS, making the first attempt to substitute the noisy Gaussian prior in established diffusion-based TTS methods with a clean and deterministic one, which provides strong structural information of the target. Specifically, we leverage the latent representation obtained from text input as our prior, and build a fully tractable Schrodinger bridge between it and the ground-truth mel-spectrogram, leading to a data-to-data process. Moreover, the tractability and flexibility of our formulation allow us to empirically study the design spaces such as noise schedules, as well as to develop stochastic and deterministic samplers. Experimental results on the LJ-Speech dataset illustrate the effectiveness of our method in terms of both synthesis quality and sampling efficiency, significantly outperforming our diffusion counterpart Grad-TTS in 50-step/1000-step synthesis and strong fast TTS models in few-step scenarios. Project page (anonymous): https://bridge-tts.github.io/.

## 1 INTRODUCTION

Diffusion models, including score-based generative models (SGMs) (Song et al., 2021b) and denoising diffusion probabilistic models (Ho et al., 2020), have been one of the most powerful generative models across different data generation tasks (Ramesh et al., 2022; Leng et al., 2022; Bao et al., 2023; Wang et al., 2023). In speech community, they have been extensively studied in waveform synthesis (Kong et al., 2021; Chen et al., 2021; 2022b), text-to-audio generation (Liu et al., 2023b;c; Huang et al., 2023b;a), and text-to-speech (TTS) synthesis (Tan et al., 2021; Popov et al., 2021; Shen et al., 2023). Generally, these models contain two processes between the data distribution and the prior distribution: 1) the forward diffusion process gradually transforms the data into a known prior distribution, e.g., Gaussian noise; 2) the reverse denoising process gradually generates data samples from the prior distribution.

In diffusion-based TTS systems (Popov et al., 2021; Chen et al., 2023; Ye et al., 2023), the text input is usually first transformed into latent representation by a text encoder, which contains a phoneme encoder and a duration predictor, and then diffusion models are employed as a decoder to generate the mel-spectrogram conditioned on the latent. The prior distribution in these systems can be classified into two types: 1) one is using the standard Gaussian noise to generate target (Huang et al., 2022; Liu et al., 2022b; Chen et al., 2022c); 2) the other improves the prior to be more informative of the target. For example, Grad-TTS (Popov et al., 2021) learns the latent representation from the text encoder with the ground-truth target in training, and takes it as the mean of prior distribution to obtain a mean-shifted Gaussian. PriorGrad (Lee et al., 2022) utilizes the statistical values from training data, computing a Gaussian with covariance matrix. DiffSinger (Liu et al., 2022a) employs an auxiliary model to acquire an intractable prior distribution, enabling a shallow reverse process. However, because diffusion models pre-specify the noise-additive diffusion process, the prior distribution of the above systems is confined to a noisy representation, which is not indicative of the mel-spectrogram.

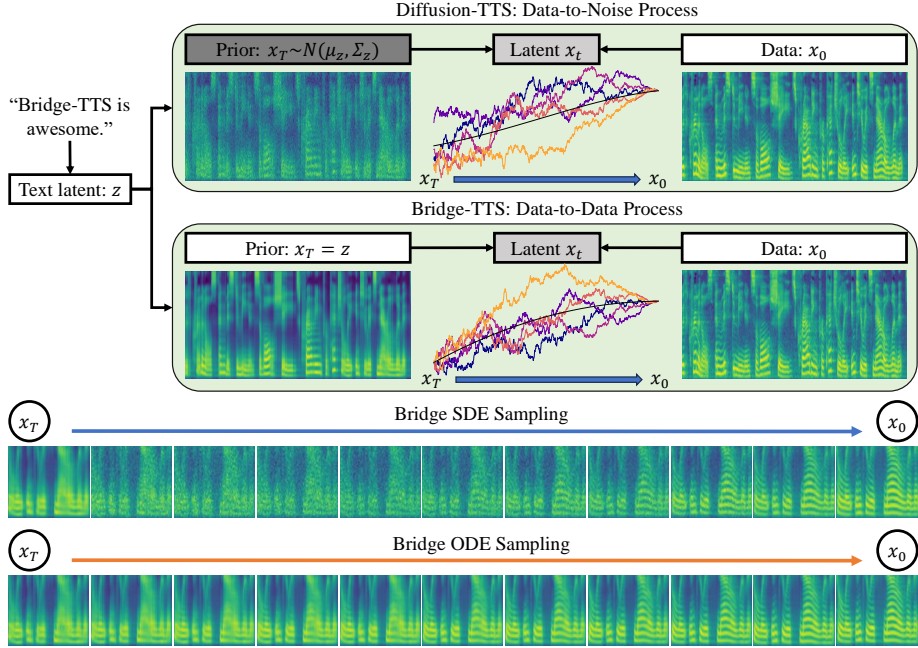

Figure 1: An overview of Bridge-TTS built on Schrodinger bridge.

In this work, as shown in Figure 1, we propose a new design to generate mel-spectrogram from a clean and deterministic prior, *i.e.*, the text latent representation supervised by ground-truth target (Popov et al., 2021). It has provided structural information of the target and is utilized as the condition information in both recent diffusion (Chen et al., 2023; Ye et al., 2023) and conditional flow matching (Guo et al., 2023; Mehta et al., 2023) based TTS systems, while we argue that replacing the noisy prior in previous systems with this clean latent can further boost the TTS sample quality and inference speed. To enable this design, we leverage Schrodinger bridge (Schrödinger, 1932; Chen et al., 2022a) instead of diffusion models, which seeks a data-to-data process rather than the data-to-noise process in diffusion models. As the original Schrodinger bridge is generally intractable that hinders the study of the design spaces in training and sampling, we propose a fully tractable Schrodinger bridge between paired data with a flexible form of reference SDE in alignment with diffusion models (Ho et al., 2020; Song et al., 2021b).

With the tractability and flexibility of our proposed framework, aiming at TTS synthesis with superior generation quality and efficient sampling speed, we make an investigation of noise schedule, model parameterization, and training-free samplers, which diffusion models have greatly benefited from (Hoogeboom et al., 2023; Salimans & Ho, 2022; Song et al., 2021a), while not been thoroughly studied in Schrodinger bridge related works. To summarize, we make the following key contributions in this work:

- In TTS synthesis, we make the first attempt to generate the mel-spectrogram from clean text latent representation (*i.e.*, the condition information in diffusion counterpart) by means of Schrodinger bridge, exploring data-to-data process rather than data-to-noise process.

- By proposing a fully tractable Schrodinger bridge between paired data with a flexible form of reference SDE, we theoretically elucidate and empirically explore the design spaces of noise schedule, model parameterization, and sampling process, further enhancing TTS quality with asymmetric noise schedule, data prediction, and first-order bridge samplers.

- Empirically, we attain both state-of-the-art generation quality and inference speed with a single training session. In both 1000-step and 50-step generation, we significantly outperform our diffusion counterpart Grad-TTS (Popov et al., 2021); in 4-step generation, we accomplish higher quality than FastGrad-TTS (Vovk et al., 2022); in 2-step generation, we surpass the state-of-the-art distillation method CoMoSpeech Ye et al. (2023), and the transformer-based model FastSpeech 2 (Ren et al., 2021).

## 2 BACKGROUND

### 2.1 DIFFUSION MODELS

Given a data distribution $p_{\text{data}}(\boldsymbol{x})$, $\boldsymbol{x} \in \mathbb{R}^d$, SGMs (Song et al., 2021b) are built on a continuous-time diffusion process defined by a forward stochastic differential equation (SDE):

$$\mathrm{d}\boldsymbol{x}_t = \boldsymbol{f}(\boldsymbol{x}_t, t)\mathrm{d}t + g(t)\mathrm{d}\boldsymbol{w}_t, \quad \boldsymbol{x}_0 \sim p_0 = p_{\text{data}} \tag{1}$$

where $t \in [0, T]$ for some finite horizon $T$, $\boldsymbol{f} : \mathbb{R}^d \times [0, T] \to \mathbb{R}^d$ is a vector-valued drift term, $g : [0, T] \to \mathbb{R}$ is a scalar-valued diffusion term, and $\boldsymbol{w}_t \in \mathbb{R}^d$ is a standard Wiener process. Under proper construction of $\boldsymbol{f}, g$, the boundary distribution $p_T(\boldsymbol{x}_T)$ is approximately a Gaussian prior distribution $p_{\text{prior}} = \mathcal{N}(\boldsymbol{0}, \sigma_T^2 \boldsymbol{I})$. The forward SDE has a corresponding reverse SDE (Song et al., 2021b) which shares the same marginal distributions $\{p_t\}_{t=0}^T$ with the forward SDE:

$$\mathrm{d}\boldsymbol{x}_t = [\boldsymbol{f}(\boldsymbol{x}_t, t) - g^2(t)\nabla \log p_t(\boldsymbol{x}_t)]\mathrm{d}t + g(t)\mathrm{d}\bar{\boldsymbol{w}}_t, \quad \boldsymbol{x}_T \sim p_T \approx p_{\text{prior}} \tag{2}$$

where $\bar{\boldsymbol{w}}_t$ is the reverse-time Wiener process, and the only unknown term $\nabla \log p_t(\boldsymbol{x}_t)$ is the *score function* of the marginal density $p_t$. By parameterizing a score network $\boldsymbol{s}_\theta(\boldsymbol{x}_t, t)$ to predict $\nabla \log p_t(\boldsymbol{x}_t)$, we can replace the true score in Eqn. (2) and solve it reversely from $p_{\text{prior}}$ at $t = T$, yielding generated data samples at $t = 0$. $\boldsymbol{s}_\theta(\boldsymbol{x}_t, t)$ is usually learned by the denoising score matching (DSM) objective (Vincent, 2011; Song et al., 2021b) with a weighting function $\lambda(t) > 0$:

$$\mathbb{E}_{p_0(\boldsymbol{x}_0)p_{t|0}(\boldsymbol{x}_t|\boldsymbol{x}_0)}\mathbb{E}_t \left[ \lambda(t)\|s_\theta(\boldsymbol{x}_t, t) - \nabla \log p_{t|0}(\boldsymbol{x}_t|\boldsymbol{x}_0)\|_2^2 \right], \tag{3}$$

where $t \sim \mathcal{U}(0, T)$ and $p_{t|0}$ is the conditional transition distribution from $\boldsymbol{x}_0$ to $\boldsymbol{x}_t$, which is determined by the pre-defined forward SDE and is analytical for a linear drift $\boldsymbol{f}(\boldsymbol{x}_t, t) = f(t)\boldsymbol{x}_t$.

### 2.2 DIFFUSION-BASED TTS SYSTEMS

The goal of TTS systems is to learn a generative model $p_\theta(\boldsymbol{x}|y)$ over mel-spectrograms (Mel) $\boldsymbol{x} \in \mathbb{R}^d$ given conditioning text $y_{1:L}$ with length $L$. Grad-TTS (Popov et al., 2021) provides a strong baseline for TTS with SGM, which consists of a text encoder and a diffusion-based decoder. Specifically, they alter the Gaussian prior in SGMs to another one $\tilde{p}_{\text{enc}}(\boldsymbol{z}|y) = \mathcal{N}(\boldsymbol{z}, \boldsymbol{I})$ with informative mean $\boldsymbol{z}$, where $\boldsymbol{z} \in \mathbb{R}^d$ is a latent acoustic feature transformed from a text string $y$ through the text encoder network $\mathcal{E}$, i.e., $\boldsymbol{z} = \mathcal{E}(y)$. The diffusion-based decoder utilizes $\tilde{p}_{\text{enc}}$ as prior for SGM and builds a diffusion process via the following modified forward SDE:

$$\mathrm{d}\boldsymbol{x}_t = \tfrac{1}{2}(\boldsymbol{z} - \boldsymbol{x}_t)\beta_t\mathrm{d}t + \sqrt{\beta_t}\mathrm{d}\boldsymbol{w}_t, \quad \boldsymbol{x}_0 \sim p_0 = p_{\text{data}}(\boldsymbol{x}|y) \tag{4}$$

where $p_0 = p_{\text{data}}(\boldsymbol{x}|y)$ is the true conditional data distribution and $\beta_t$ is a non-negative noise schedule. The forward SDE in Eqn. (4) will yield $\boldsymbol{x}_T \sim p_T \approx \tilde{p}_{\text{enc}}$ with sufficient large $T$ (Popov et al., 2021). During training, the text encoder and the diffusion-based decoder are jointly optimized, where the encoder is optimized with a negative log-likelihood loss $\mathcal{L}_{\text{enc}} = -\mathbb{E}_{p_{\text{data}}(\boldsymbol{x}|y)}[\log \tilde{p}_{\text{enc}}(\boldsymbol{x}|y)]$ and the decoder is trained with the DSM objective in Eqn. (3), denoted as $\mathcal{L}_{\text{diff}}$. Apart from $\mathcal{L}_{\text{enc}}$ and $\mathcal{L}_{\text{diff}}$, the TTS system also optimizes a duration predictor $\hat{A}$ as a part of the encoder that predicts the alignment map $A^*$ between encoded text sequence $\tilde{\boldsymbol{z}}_{1:L}$ and the latent feature $\boldsymbol{z}_{1:F}$ with $F$ frames given by Monotonic Alignment Search (Kim et al., 2020), where $\boldsymbol{z}_j = \tilde{\boldsymbol{z}}_{A^*(j)}$. Denote the duration prediction loss as $\mathcal{L}_{\text{dp}}$, the overall training objective of Grad-TTS is $\mathcal{L}_{\text{grad-tts}} = \mathcal{L}_{\text{enc}} + \mathcal{L}_{\text{dp}} + \mathcal{L}_{\text{diff}}$.

### 2.3 SCHRODINGER BRIDGE

The Schrodinger Bridge (SB) problem (Schrödinger, 1932; De Bortoli et al., 2021; Chen et al., 2022a) originates from the optimization of path measures with constrained boundaries:

$$\min_{p \in \mathcal{P}_{[0,T]}} D_{\text{KL}}(p \parallel p^{\text{ref}}), \quad s.t. \ p_0 = p_{\text{data}}, p_T = p_{\text{prior}} \tag{5}$$

where $\mathcal{P}_{[0,T]}$ is the space of path measures on a finite time horizon $[0, T]$, $p^{\text{ref}}$ is the *reference path measure*, and $p_0, p_T$ are the marginal distributions of $p$ at boundaries. Generally, $p^{\text{ref}}$ is defined by the same form of forward SDE as SGMs in Eqn. (1) (i.e., the *reference SDE*). In such a case, the

SB problem is equivalent to a couple of forward-backward SDEs (Wang et al., 2021; Chen et al., 2022a):

$$\mathrm{d}\boldsymbol{x}_t = [\boldsymbol{f}(\boldsymbol{x}_t, t) + g^2(t)\nabla \log \Psi_t(\boldsymbol{x}_t)]\mathrm{d}t + g(t)\mathrm{d}\boldsymbol{w}_t, \quad \boldsymbol{x}_0 \sim p_{\text{data}} \tag{6a}$$

$$\mathrm{d}\boldsymbol{x}_t = [\boldsymbol{f}(\boldsymbol{x}_t, t) - g^2(t)\nabla \log \widehat{\Psi}_t(\boldsymbol{x}_t)]\mathrm{d}t + g(t)\mathrm{d}\bar{\boldsymbol{w}}_t, \quad \boldsymbol{x}_T \sim p_{\text{prior}} \tag{6b}$$

where $\boldsymbol{f}$ and $g$ are the same as in the reference SDE. The extra non-linear drift terms $\nabla \log \Psi_t(\boldsymbol{x}_t)$ and $\nabla \log \widehat{\Psi}_t(\boldsymbol{x}_t)$ are also described by the following coupled partial differential equations (PDEs):

$$\begin{cases} \frac{\partial \Psi}{\partial t} = -\nabla_{\boldsymbol{x}}\Psi^\top \boldsymbol{f} - \frac{1}{2}\operatorname{Tr}\left(g^2 \nabla_{\boldsymbol{x}}^2 \Psi\right) \\ \frac{\partial \widehat{\Psi}}{\partial t} = -\nabla_{\boldsymbol{x}} \cdot (\widehat{\Psi}\boldsymbol{f}) + \frac{1}{2}\operatorname{Tr}\left(g^2 \nabla_{\boldsymbol{x}}^2 \widehat{\Psi}\right) \end{cases} \quad s.t. \ \Psi_0\widehat{\Psi}_0 = p_{\text{data}}, \Psi_T\widehat{\Psi}_T = p_{\text{prior}}. \tag{7}$$

The marginal distribution $p_t$ of the SB at any time $t \in [0, T]$ satisfies $p_t = \Psi_t\widehat{\Psi}_t$. Compared to SGMs where $p_T \approx p_{\text{prior}} = \mathcal{N}(\boldsymbol{\mu}, \sigma_T^2\boldsymbol{I})$, SB allows for a flexible form of $p_{\text{prior}}$ and ensures the boundary condition $p_T = p_{\text{prior}}$. However, solving the SB requires simulating stochastic processes and performing costly iterative procedures (De Bortoli et al., 2021; Chen et al., 2022a; Shi et al., 2023). Therefore, it suffers from scalability and applicability issues. In certain scenarios, such as using paired data as boundaries, the SB problem can be solved in a simulation-free approach (Somnath et al., 2023; Liu et al., 2023a). Nevertheless, SBs in these works are either not fully tractable or limited to restricted families of $p^{\text{ref}}$, thus lacking a comprehensive and theoretical analysis of the design spaces.

## 3 BRIDGE-TTS

We extend SB techniques to the TTS task and elucidate the design spaces with theory-grounded analyses. We start with a fully tractable SB between paired data in TTS modeling. Based on such formulation, we derive different training objectives and theoretically study SB sampling in the form of SDE and ODE, which lead to novel first-order sampling schemes when combined with exponential integrators. In the following discussions, we say two probability density functions are the same when they are up to a normalizing factor. Besides, we assume the maximum time $T = 1$ for convenience.

### 3.1 SCHRODINGER BRIDGE BETWEEN PAIRED DATA

As we have discussed, with the properties of unrestricted prior form and strict boundary condition, SB is a natural substitution for diffusion models when we have a strong informative prior. In the TTS task, the pairs of the ground-truth data $(\boldsymbol{x}, y)$ and the deterministic prior $\boldsymbol{z} = \mathcal{E}(y)$ given by the text encoder can be seen as mixtures of dual Dirac distribution boundaries $(\delta_{\boldsymbol{x}}, \delta_{\boldsymbol{z}})$, which simplifies the solving of SB problem. However, in such a case, the SB problem in Eqn. (5) will inevitably collapse given a stochastic reference process that admits a continuous density $p_1^{\text{ref}}$ at $t = 1$, since the KL divergence between a Dirac distribution and a continuous probability measure is infinity.

To tackle this problem, we consider a noisy observation of boundary data points $\boldsymbol{x}_0, \boldsymbol{x}_1$ polluted by a small amount of Gaussian noise $\mathcal{N}(\boldsymbol{0}, \epsilon_1^2\boldsymbol{I})$ and $\mathcal{N}(\boldsymbol{0}, \epsilon_2^2\boldsymbol{I})$ respectively, which helps us to identify the SB formulation between clean data when $\epsilon_1, \epsilon_2 \to 0$. Actually, we show that in general cases where the reference SDE has a linear drift $\boldsymbol{f}(\boldsymbol{x}_t, t) = f(t)\boldsymbol{x}_t$ (which is aligned with SGMs), SB has a fully tractable and neat solution when $\epsilon_2 = e^{\int_0^1 f(\tau)\mathrm{d}\tau}\epsilon_1$. We formulate the result in the following theorem.

**Proposition 3.1** (Tractable Schrodinger Bridge between Gaussian-Smoothed Paired Data with Reference SDE of Linear Drift, proof in Appendix A.1). *Assume $\boldsymbol{f} = f(t)\boldsymbol{x}_t$, the analytical solution to Eqn. (7) when $p_{data} = \mathcal{N}(\boldsymbol{x}_0, \epsilon^2\boldsymbol{I})$ and $p_{prior} = \mathcal{N}(\boldsymbol{x}_1, e^{2\int_0^1 f(\tau)\mathrm{d}\tau}\epsilon^2\boldsymbol{I})$ is*

$$\widehat{\Psi}_t^\epsilon = \mathcal{N}(\alpha_t\boldsymbol{a}, (\alpha_t^2\sigma^2 + \alpha_t^2\sigma_t^2)\boldsymbol{I}), \quad \Psi_t^\epsilon = \mathcal{N}(\bar{\alpha}_t\boldsymbol{b}, (\alpha_t^2\sigma^2 + \alpha_t^2\bar{\sigma}_t^2)\boldsymbol{I}) \tag{8}$$

*where $t \in [0, 1]$,*

$$\boldsymbol{a} = \boldsymbol{x}_0 + \frac{\sigma^2}{\sigma_1^2}(\boldsymbol{x}_0 - \frac{\boldsymbol{x}_1}{\alpha_1}), \quad \boldsymbol{b} = \boldsymbol{x}_1 + \frac{\sigma^2}{\sigma_1^2}(\boldsymbol{x}_1 - \alpha_1\boldsymbol{x}_0), \quad \sigma^2 = \epsilon^2 + \frac{\sqrt{\sigma_1^4 + 4\epsilon^4} - \sigma_1^2}{2}, \tag{9}$$

*and*

$$\alpha_t = e^{\int_0^t f(\tau)\mathrm{d}\tau}, \quad \bar{\alpha}_t = e^{-\int_t^1 f(\tau)\mathrm{d}\tau}, \quad \sigma_t^2 = \int_0^t \frac{g^2(\tau)}{\alpha_\tau^2}\mathrm{d}\tau, \quad \bar{\sigma}_t^2 = \int_t^1 \frac{g^2(\tau)}{\alpha_\tau^2}\mathrm{d}\tau. \tag{10}$$

In the above theorem, $\alpha_t, \bar{\alpha}_t, \sigma_t, \bar{\sigma}_t$ are determined by $f, g$ in the reference SDE (Eqn. (1)) and are analogous to the *noise schedule* in SGMs (Kingma et al., 2021). When $\epsilon \to 0$, $\widehat{\Psi}_t^\epsilon, \Psi_t^\epsilon$ converge to the tractable solution between clean paired data $(\boldsymbol{x}_0, \boldsymbol{x}_1)$:

$$\widehat{\Psi}_t = \mathcal{N}(\alpha_t \boldsymbol{x}_0, \alpha_t^2 \sigma_t^2 \boldsymbol{I}), \quad \Psi_t = \mathcal{N}(\bar{\alpha}_t \boldsymbol{x}_1, \alpha_t^2 \bar{\sigma}_t^2 \boldsymbol{I}) \tag{11}$$

The advantage of such tractability lies in its ability to facilitate the study of training and sampling under the forward-backward SDEs (Eqn. (6)), which we will discuss in the following sections. Besides, the marginal distribution $p_t = \widehat{\Psi}_t \Psi_t$ of the SB also has a tractable form:

$$p_t = \Psi_t \widehat{\Psi}_t = \mathcal{N}\left( \frac{\alpha_t \bar{\sigma}_t^2 \boldsymbol{x}_0 + \bar{\alpha}_t \sigma_t^2 \boldsymbol{x}_1}{\sigma_1^2}, \frac{\alpha_t^2 \bar{\sigma}_t^2 \sigma_t^2}{\sigma_1^2} \boldsymbol{I} \right), \tag{12}$$

which is a Gaussian distribution whose mean is an interpolation between $\boldsymbol{x}_0, \boldsymbol{x}_1$, and variance is zero at boundaries and positive at the middle. A special case is that, when the noise schedule $f(t) = 0$ and $g(t) = \sigma > 0$, we have $p_t = \mathcal{N}((1-t)\boldsymbol{x}_0 + t\boldsymbol{x}_1, \sigma^2 t(1-t)\boldsymbol{I})$, which recovers the Brownian bridge used in previous works (Qiu et al., 2023; Tong et al., 2023a;b). Actually, Eqn. (12) reveals the form of *generalized Brownian bridge with linear drift and time-varying volatility* between $\boldsymbol{x}_0$ and $\boldsymbol{x}_1$. We put the detailed analysis in Appendix B.1.

## 3.2 Model Training

The TTS task aims to learn a model to generate the Mel $\boldsymbol{x}_0$ given text $y$. Denote $\boldsymbol{x}_1 = \mathcal{E}(y)$ as the latent acoustic feature produced by text encoder $\mathcal{E}$, since the SB is tractable given $\boldsymbol{x}_0, \boldsymbol{x}_1$ ($\nabla \log \Psi, \nabla \log \widehat{\Psi}$ in Eqn. (6) are determined by Eqn. (11)), a direct training approach is to parameterize a network $\boldsymbol{x}_\theta$ to predict $\boldsymbol{x}_0$ given $\boldsymbol{x}_t$ at different timesteps, which allows us to simulate the process of SB from $t = 1$ to $t = 0$. This is in alignment with the *data prediction* in diffusion models, and we have the bridge loss:

$$\mathcal{L}_{\text{bridge}} = \mathbb{E}_{(\boldsymbol{x}_0, y) \sim p_{\text{data}}, \boldsymbol{x}_1 = \mathcal{E}(y)} \mathbb{E}_t [\|\boldsymbol{x}_\theta(\boldsymbol{x}_t, t, \boldsymbol{x}_1) - \boldsymbol{x}_0\|_2^2] \tag{13}$$

where $\boldsymbol{x}_t = \frac{\alpha_t \bar{\sigma}_t^2}{\sigma_1^2} \boldsymbol{x}_0 + \frac{\bar{\alpha}_t \sigma_t^2}{\sigma_1^2} \boldsymbol{x}_1 + \frac{\alpha_t \bar{\sigma}_t \sigma_t}{\sigma_1} \boldsymbol{\epsilon}, \boldsymbol{\epsilon} \sim \mathcal{N}(\boldsymbol{0}, \boldsymbol{I})$ by the SB (Eqn. (12)). $\boldsymbol{x}_1$ is also fed into the network as a condition, following Grad-TTS (Popov et al., 2021).

Analogous to the different parameterizations in diffusion models, there are alternative choices of training objectives that are equivalent in bridge training, such as the noise prediction corresponding to $\nabla \log \widehat{\Psi}_t$ (Liu et al., 2023a) or the SB score $\nabla \log p_t$, and the velocity prediction related to flow matching techniques (Tong et al., 2023a;b). However, we find they perform worse or poorly in practice, which we will discuss in detail in Appendix D. Except for the bridge loss, we jointly train the text encoder $\mathcal{E}$ (including the duration predictor $\hat{A}$) following Grad-TTS. Since the encoder no longer parameterizes a Gaussian distribution, we simply adopt an MSE encoder loss $\mathcal{L}'_{\text{enc}} = \mathbb{E}_{(\boldsymbol{x}_0, y) \sim p_{\text{data}}} \|\mathcal{E}(y) - \boldsymbol{x}_0\|^2$. And we use the same duration prediction loss $\mathcal{L}_{\text{dp}}$ as Grad-TTS. The overall training objective of Bridge-TTS is $\mathcal{L}_{\text{bridge-tts}} = \mathcal{L}'_{\text{enc}} + \mathcal{L}_{\text{dp}} + \mathcal{L}_{\text{bridge}}$.

In our framework, the flexible form of reference SDE facilitates the design of noise schedules $f, g$, which constitutes an important factor of performance as in SGMs. In this work, we directly transfer the well-behaved noise schedules from SGMs, such as variance preserving (VP). As shown in Table 1, we set $f, g^2$ linear to $t$, and the corresponding $\alpha_t, \sigma_t^2$ have closed-form expressions. Such designs are new in both SB and TTS-related contexts and distinguish our work from previous ones with Brownian bridges (Qiu et al., 2023; Tong et al., 2023a;b).

Table 1: Demonstration of the noise schedules in Bridge-TTS.

| Schedule | $f(t)$ | $g^2(t)$ | $\alpha_t$ | $\sigma_t^2$ |
|---|---|---|---|---|
| Bridge-gmax[a] | $0$ | $\beta_0 + t(\beta_1 - \beta_0)$ | $1$ | $\frac{1}{2}(\beta_1 - \beta_0)t^2 + \beta_0 t$ |
| Bridge-VP | $-\frac{1}{2}(\beta_0 + t(\beta_1 - \beta_0))$ | $\beta_0 + t(\beta_1 - \beta_0)$ | $e^{-\frac{1}{2}\int_0^t (\beta_0 + \tau(\beta_1 - \beta_0))\mathrm{d}\tau}$ | $e^{\int_0^t (\beta_0 + \tau(\beta_1 - \beta_0))\mathrm{d}\tau} - 1$ |

[a]The main hyperparameter for the Bridge-gmax schedule is $\beta_1$, which is exactly the maximum of $g^2(t)$.

### 3.3 SAMPLING SCHEME

Assume we have a trained data prediction network $\boldsymbol{x}_\theta(\boldsymbol{x}_t, t)$ [1]. If we replace $\boldsymbol{x}_0$ with $\boldsymbol{x}_\theta$ in the tractable solution of $\widehat{\Psi}, \Psi$ (Eqn. (11)) and substitute them into Eqn. (6), which describes the SB with SDEs, we can obtain the parameterized SB process. Analogous to the sampling in diffusion models, the parameterized SB can be described by both stochastic and deterministic processes, which we call bridge SDE/ODE, respectively.

**Bridge SDE** We can follow the reverse SDE in Eqn. (6b). By substituting Eqn. (11) into it and replace $\boldsymbol{x}_0$ with $\boldsymbol{x}_\theta$, we have the *bridge SDE*:

$$\mathrm{d}\boldsymbol{x}_t = \left[ f(t)\boldsymbol{x}_t + g^2(t)\frac{\boldsymbol{x}_t - \alpha_t \boldsymbol{x}_\theta(\boldsymbol{x}_t, t)}{\alpha_t^2 \sigma_t^2} \right] \mathrm{d}t + g(t)\mathrm{d}\bar{\boldsymbol{w}}_t \tag{14}$$

**Bridge ODE** The *probability flow ODE* (Song et al., 2021b) of the forward SDE in Eqn. (6a) is (Chen et al., 2022a):

$$
\begin{aligned}
\mathrm{d}\boldsymbol{x}_t &= \left[ \boldsymbol{f}(t)\boldsymbol{x}_t + g^2(t)\nabla \log \Psi_t(\boldsymbol{x}_t) - \tfrac{1}{2}g^2(t)\nabla \log p_t(\boldsymbol{x}_t) \right] \mathrm{d}t \\
&= \left[ \boldsymbol{f}(t)\boldsymbol{x}_t + \tfrac{1}{2}g^2(t)\nabla \log \Psi_t(\boldsymbol{x}_t) - \tfrac{1}{2}g^2(t)\nabla \log \widehat{\Psi}_t(\boldsymbol{x}_t) \right] \mathrm{d}t
\end{aligned}
\tag{15}
$$

where we have used $\nabla \log p_t(\boldsymbol{x}_t) = \nabla \log \Psi_t(\boldsymbol{x}_t) + \nabla \log \widehat{\Psi}_t(\boldsymbol{x}_t)$ since $p_t = \Psi_t \widehat{\Psi}_t$. By substituting Eqn. (11) into it and replace $\boldsymbol{x}_0$ with $\boldsymbol{x}_\theta$, we have the *bridge ODE*:

$$\mathrm{d}\boldsymbol{x}_t = \left[ \boldsymbol{f}(t)\boldsymbol{x}_t - \tfrac{1}{2}g^2(t)\frac{\boldsymbol{x}_t - \bar{\alpha}_t \boldsymbol{x}_1}{\alpha_t^2 \bar{\sigma}_t^2} + \tfrac{1}{2}g^2(t)\frac{\boldsymbol{x}_t - \alpha_t \boldsymbol{x}_\theta(\boldsymbol{x}_t, t)}{\alpha_t^2 \sigma_t^2} \right] \mathrm{d}t \tag{16}$$

To obtain data sample $\boldsymbol{x}_0$, we can solve the bridge SDE/ODE from the latent $\boldsymbol{x}_1$ at $t = 1$ to $t = 0$. However, directly solving the bridge SDE/ODE may cause large errors when the number of steps is small. A prevalent technique in diffusion models is to handle them with exponential integrators (Lu et al., 2022a;b; Gonzalez et al., 2023), which aims to "cancel" the linear terms involving $\boldsymbol{x}_t$ and obtain solutions with lower discretization error. We conduct similar derivations for bridge sampling, and present the results in the following theorem.

**Proposition 3.2** (Exact Solution and First-Order Discretization of Bridge SDE/ODE, proof in Appendix A.2)**.** *Given an initial value $\boldsymbol{x}_s$ at time $s > 0$, the solution at time $t \in [0, s]$ of bridge SDE/ODE is*

$$\boldsymbol{x}_t = \frac{\alpha_t \sigma_t^2}{\alpha_s \sigma_s^2}\boldsymbol{x}_s - \alpha_t \sigma_t^2 \int_s^t \frac{g^2(\tau)}{\alpha_\tau^2 \sigma_\tau^4}\boldsymbol{x}_\theta(\boldsymbol{x}_\tau, \tau)\mathrm{d}\tau + \alpha_t \sigma_t \sqrt{1 - \frac{\sigma_t^2}{\sigma_s^2}}\boldsymbol{\epsilon}, \quad \boldsymbol{\epsilon} \sim \mathcal{N}(\boldsymbol{0}, \boldsymbol{I}) \tag{17}$$

$$\boldsymbol{x}_t = \frac{\alpha_t \sigma_t \bar{\sigma}_t}{\alpha_s \sigma_s \bar{\sigma}_s}\boldsymbol{x}_s + \frac{\bar{\alpha}_t \sigma_t^2}{\sigma_1^2}\left( 1 - \frac{\sigma_s \bar{\sigma}_t}{\bar{\sigma}_s \sigma_t} \right)\boldsymbol{x}_1 - \frac{\alpha_t \sigma_t \bar{\sigma}_t}{2}\int_s^t \frac{g^2(\tau)}{\alpha_\tau^2 \sigma_\tau^3 \bar{\sigma}_\tau}\boldsymbol{x}_\theta(\boldsymbol{x}_\tau, \tau)\mathrm{d}\tau \tag{18}$$

*The first-order discretization (with the approximation $\boldsymbol{x}_\theta(\boldsymbol{x}_\tau, \tau) \approx \boldsymbol{x}_\theta(\boldsymbol{x}_s, s)$ for $\tau \in [t, s]$) gives*

$$\boldsymbol{x}_t = \frac{\alpha_t \sigma_t^2}{\alpha_s \sigma_s^2}\boldsymbol{x}_s + \alpha_t\left( 1 - \frac{\sigma_t^2}{\sigma_s^2} \right)\boldsymbol{x}_\theta(\boldsymbol{x}_s, s) + \alpha_t \sigma_t \sqrt{1 - \frac{\sigma_t^2}{\sigma_s^2}}\boldsymbol{\epsilon}, \quad \boldsymbol{\epsilon} \sim \mathcal{N}(\boldsymbol{0}, \boldsymbol{I}) \tag{19}$$

$$\boldsymbol{x}_t = \frac{\alpha_t \sigma_t \bar{\sigma}_t}{\alpha_s \sigma_s \bar{\sigma}_s}\boldsymbol{x}_s + \frac{\alpha_t}{\sigma_1^2}\left[ \left( \bar{\sigma}_t^2 - \frac{\bar{\sigma}_s \sigma_t \bar{\sigma}_t}{\sigma_s} \right)\boldsymbol{x}_\theta(\boldsymbol{x}_s, s) + \left( \sigma_t^2 - \frac{\sigma_s \sigma_t \bar{\sigma}_t}{\bar{\sigma}_s} \right)\frac{\boldsymbol{x}_1}{\alpha_1} \right] \tag{20}$$

To the best of our knowledge, such derivations are revealed for the first time in the context of SB. We find that the first-order discretization of bridge SDE (Eqn. (19)) recovers posterior sampling (Liu et al., 2023a) on a Brownian bridge, and the first-order discretization of bridge ODE (Eqn. (20)) in the limit of $\frac{\sigma_s}{\sigma_1}, \frac{\sigma_t}{\sigma_1} \to 0$ recovers deterministic DDIM sampler (Song et al., 2021a) in diffusion models. Besides, we can easily discover that the 1-step case of Eqn. (19) and Eqn. (20) are both 1-step deterministic prediction by $\boldsymbol{x}_\theta$. We put more detailed analyses in Appendix B.2.

We can also develop higher-order samplers by taking higher-order Taylor expansions for $\boldsymbol{x}_\theta$ in the exact solutions. We further discuss and take the predictor-corrector method as the second-order case in Appendix C. In practice, we find first-order sampler is enough for the TTS task, and higher-order samplers do not make any significant difference.

---

[1] We omit the condition $\boldsymbol{x}_1$ for simplicity and other parameterizations such as noise prediction can be first transformed to $\boldsymbol{x}_\theta$.

## 4 EXPERIMENTS

### 4.1 TRAINING SETUP

**Data**: We utilize the LJ-Speech dataset (Ito & Johnson, 2017), which contains $13,100$ samples, around 24 hours in total, from a female speaker at a sampling rate of 22.05 kHz. The test samples are extracted from both LJ-001 and LJ-002, and the remaining 12577 samples are used for training. We follow the common practice, using the open-source tools (Park, 2019) to convert the English grapheme sequence to phoneme sequence, and extracting the 80-band mel-spectrogram with the FFT 1024 points, 80Hz and 7600Hz lower and higher frequency cutoffs, and a hop length of 256.

**Model training**: To conduct a fair comparison with diffusion models, we adopt the same network architecture and training settings used in Grad-TTS (Popov et al., 2021): 1) the encoder (*i.e.*, text encoder and duration predictor) contains 7.2M parameters and the U-Net based decoder contains 7.6M parameters; 2) the model is trained with a batch size of 16, and 1.7M iterations in total on a single NVIDIA RTX 3090, using 2.5 days; 3) the Adam optimizer (Kingma & Ba, 2015) is employed with a constant learning rate of 0.0001. For noise schedules, we set $\beta_0 = 0.01, \beta_1 = 20$ for Bridge-VP (exactly the same as VP in SGMs) and $\beta_0 = 0.01, \beta_1 = 50$ for Bridge-gmax.

**Evaluation**: Following previous works (Popov et al., 2021; Liu et al., 2022a; Huang et al., 2022), we conduct the subjective tests MOS (Mean Opinion Score) and CMOS (Comparison Mean Opinion Score) to evaluate the overall subjective quality and comparison sample quality, respectively. To guarantee the reliability of the collected results, we use the open platform *Amazon Mechanical Turk*, and require Master workers to complete the listening test. Specifically, the MOS scores of 20 test samples are given by 25 Master workers to evaluate the overall performance with a 5-point scale, where 1 and 5 denote the lowest ("Bad") and highest ("Excellent") quality respectively. The result is reported with a 95% confidence interval. Each CMOS score is given by 15 Master workers to compare 20 test samples synthesized by two different models. Each of the test samples has been normalized for a fair comparison[2]. To measure the inference speed, we calculate the real-time factor (RTF) on an NVIDIA RTX 3090.

Table 2: The MOS comparison with 95% confidence interval given numerous sampling steps.

| Model | NFE | RTF ($\downarrow$) | MOS ($\uparrow$) |
|---|---|---|---|
| Recording | / | / | $4.10 \pm 0.06$ |
| GT-Mel + voc. | / | / | $3.93 \pm 0.07$ |
| FastSpeech 2 | 1 | 0.004 | $3.78 \pm 0.07$ |
| VITS | 1 | 0.018 | $3.99 \pm 0.07$ |
| DiffSinger | 71 | 0.157 | $3.92 \pm 0.06$ |
| ResGrad | 50 | 0.135 | $3.97 \pm 0.07$ |
| Grad-TTS | 50 | 0.116 | $3.99 \pm 0.07$ |
| **Ours (VP)** | 50 | 0.117 | $\mathbf{4.09 \pm 0.07}$ |
| **Ours (gmax)** | 50 | 0.117 | $\mathbf{4.07 \pm 0.07}$ |
| Grad-TTS | 1000 | 2.233 | $3.98 \pm 0.07$ |
| **Ours (VP)** | 1000 | 2.267 | $\mathbf{4.05 \pm 0.07}$ |
| **Ours (gmax)** | 1000 | 2.267 | $\mathbf{4.07 \pm 0.07}$ |

Table 3: The MOS comparison with 95% confidence interval in few-step generation.

| Model | NFE | RTF ($\downarrow$) | MOS ($\uparrow$) |
|---|---|---|---|
| Recording | / | / | $4.12 \pm 0.06$ |
| GT-Mel + voc. | / | / | $4.01 \pm 0.06$ |
| FastSpeech 2 | 1 | 0.004 | $3.84 \pm 0.07$ |
| CoMoSpeech | 1 | 0.007 | $3.74 \pm 0.07$ |
| ProDiff | 2 | 0.019 | $3.67 \pm 0.07$ |
| CoMoSpeech | 2 | 0.009 | $3.87 \pm 0.07$ |
| **Ours (gmax)** | 2 | 0.009 | $\mathbf{4.04 \pm 0.06}$ |
| DiffGAN-TTS | 4 | 0.014 | $3.78 \pm 0.07$ |
| Grad-TTS | 4 | 0.013 | $3.88 \pm 0.07$ |
| FastGrad-TTS | 4 | 0.013 | $3.87 \pm 0.07$ |
| ResGrad | 4 | 0.017 | $4.02 \pm 0.06$ |
| **Ours (gmax)** | 4 | 0.013 | $\mathbf{4.10 \pm 0.06}$ |

### 4.2 RESULTS AND ANALYSES

We demonstrate the performance of Bridge-TTS on sample quality and inference speed separately, which guarantees a more precise comparison between multiple models. In Table 2 and Table 3, the test samples in LJ-Speech dataset are denoted as *Recording*, the samples synthesized from ground-truth mel-spectrogram by vocoder is denoted as *GT-Mel+voc.*, and the number of function evaluations is denoted as *NFE*. We take the pre-trained HiFi-GAN (Kong et al., 2020)[3] as the vocoder,

---

[2]https://github.com/slhck/ffmpeg-normalize
[3]https://github.com/jik876/hifi-gan

aligned with other baseline settings. More details of baseline models are introduced in Appendix F. In the sampling process of both tests, Grad-TTS employs ODE sampling and sets the prior distribution $p_T = \mathcal{N}(z, \tau_d^{-1}I)$ with a temperature parameter $\tau_d = 1.5$. In Bridge-TTS, we use our first-order SDE sampler shown in Eqn. (19) with a temperature parameter $\tau_b = 2$ for the noise distribution $\epsilon = \mathcal{N}(0, \tau_b^{-1}I)$, which is helpful to the TTS quality in our observation.

**Generation quality.** Table 2 compares the generation quality between Bridge-TTS and previous TTS systems. As shown, both Bridge-TTS models outperform three strong diffusion-based TTS systems: our diffusion counterpart Grad-TTS (Popov et al., 2021), the shallow diffusion model DiffSinger (Liu et al., 2022a) and the residual diffusion model ResGrad (Chen et al., 2022c). In comparison with the transformer-based model FastSpeech 2 (Ren et al., 2021) and the end-to-end TTS system (Kim et al., 2021), we also exhibit stronger subjective quality. When NFE is either 1000 or 50, our Bridge-TTS achieves superior quality. One reason is that the condition information (*i.e.*, text encoder output) in TTS synthesis is strong, and the other is that our first-order Bridger sampler maintains the sample quality when reducing the NFE.

**Sampling speed.** Table 3 shows the evaluation of sampling speed with the Bridge-TTS-gmax model, as we observe that it achieves higher quality than the VP-based Bridge-TTS system. To conduct a fair comparison, we choose the NFE reported in the baseline models. As shown, in 4-step sampling, we not only outperform our diffusion counterpart Grad-TTS (Popov et al., 2021), FastGrad-TTS (Vovk et al., 2022) using a first-order SDE sampler, and DiffGAN-TTS (Liu et al., 2022b) by a large margin, but also achieve higher quality than ResGrad (Chen et al., 2022c) which stands on a pre-trained FastSpeech 2 (Ren et al., 2021). In 2-step sampling with a RTF of 0.009, we achieve higher quality than the state-of-the-art fast sampling method CoMoSpeech (Ye et al., 2023). In comparison with 1-step method, FastSpeech 2 and CoMoSpeech, although our 2-step generation is slightly slower, we achieve distinctively better quality.

### 4.3 CASE STUDY

We show a sample when NFE=4 in Figure 2 (a), using our first-order ODE sampler shown in Eqn (20). As shown, Bridge-TTS clearly generates more details of the target than the diffusion counterpart Grad-TTS ($\tau_d = 1.5$). Moreover, we show a 2-step ODE sampling trajectory of Bridge-TTS in Figure 2 (b). As shown, with our data-to-data generation process, each sampling step is adding more details to refine the prior which has provided strong information about the target. More generated samples can be visited in Appendix G.

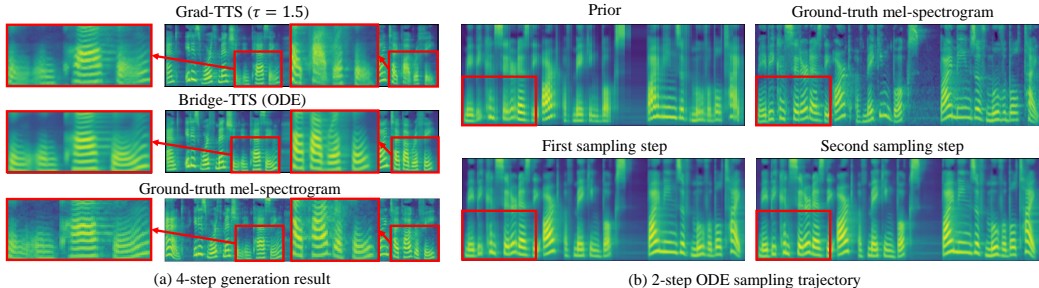

(a) 4-step generation result        (b) 2-step ODE sampling trajectory

Figure 2: We show a 4-step ODE generation result of Grad-TTS (Popov et al., 2021) and Bridge-TTS in the left figure, and a 2-step ODE sampling trajectory of Bridge-TTS in the right one. The ground-truth mel-spectrogram is shown for comparison.

### 4.4 ABLATION STUDY

We conduct several comparison studies by showing the CMOS results between different designs of prior, noise schedule, and sampler when NFE=1000 and NFE=4. The base setup is the Bridge-gmax schedule, $x_0$ predictor, and temperature-scaled first-order SDE sampler ($\tau_b = 2$).

**Prior.** We explore two training strategies that differ in their prior: 1) like Grad-TTS (Popov et al., 2021), the encoder and decoder part are joint trained from scratch (i.e., mutable prior); 2) the encoder is first trained with a warm-up stage and then the decoder is trained from scratch (i.e., fixed prior).

It should be noted that in both strategies, the text encoder is trained with an equivalent objective. As shown, the latter consistently has better sample quality across different NFEs. Hence, we adopt it as our default setting.

**Noise schedule.** We compare three different configurations for noise schedules: Bridge-gmax, Bridge-VP, and a simple schedule with $f(t) = 0, g(t) = 5$ that has virtually the same maximum marginal variance as Bridge-gmax, which we refer to as "constant $g(t)$". As shown in Table 4, Bridge-gmax and Bridge-VP have

Table 4: CMOS comparison of training and sampling settings of Bridge-TTS.

| Method | NFE=4 | NFE=1000 |
|---|---|---|
| Bridge-TTS (gmax) | 0 | 0 |
| *w.* mutable prior | - 0.13 | - 0.17 |
| *w.* constant $g(t)$ | - 0.12 | - 0.14 |
| *w.* VP | - 0.03 | - 0.08 |
| *w.* SDE ($\tau_b = 1$) | - 0.07 | - 0.19 |
| *w.* ODE | - 0.10 | + 0.00 |

overall similar performance, while the constant $g(t)$ has noticeably degraded quality than Bridge-gmax when NFE=1000. Intuitively, the Bridge-gmax and Bridge-VP have an asymmetric pattern of marginal variance that assigns more steps for denoising, while the constant $g(t)$ yields a symmetric pattern. Empirically, such an asymmetric pattern of marginal variance helps improve sample quality. We provide a more detailed illustration of the noise schedules in Appendix E.

**Sampling process.** For comparison between different sampling processes, the temperature-scaled SDE ($\tau_b = 2$) achieves the best quality at both NFE=4 and NFE=1000. Compared with the vanilla SDE sampling (i.e., $\tau_b = 1$), introducing the temperature sampling technique for SDE can effectively reduce artifacts in the background and enhance the sample quality when NFE is large, which is clearly reflected in the CMOS score in Table 4. Meanwhile, the ODE sampler exhibits the same quality as the temperature-scaled SDE at NFE=1000, but it has more evident artifacts at NFE=4.

## 5 RELATED WORK

**Diffusion-based TTS Synthesis.** Grad-TTS (Popov et al., 2021) builds a strong TTS baseline with SGMs, surpassing the transformer-based (Ren et al., 2019) and flow-based model (Kim et al., 2020). In the following works, fast sampling methods are extensively studied, such as improving prior distribution (Lee et al., 2022), designing training-free sampler (Jeong et al., 2021; Vovk et al., 2022), using auxiliary model (Liu et al., 2022a; Chen et al., 2022c), introducing adversarial loss (Liu et al., 2022b; Ko & Choi, 2023), employing knowledge distillation (Huang et al., 2022; Ye et al., 2023), developing lightweight U-Net (Chen et al., 2023), and leveraging CFM framework (Mehta et al., 2023; Guo et al., 2023; Guan et al., 2023). However, these methods usually explore to find a better trade-off between TTS quality and sampling speed than diffusion models instead of simultaneously improving both of them, and some of these methods require extra procedures, such as data pre-processing, auxiliary networks, and distillation stage, or prone to training instability. In contrast to each of the previous methods that study a data-to-noise process, we present a novel TTS system with a tractable Schrodinger bridge, demonstrating the advantages of the data-to-data process.

**Schrodinger bridge.** Solving the Schrodinger bridge problem with an iterative procedure to simulate the intractable stochastic processes is widely studied (De Bortoli et al., 2021; Wang et al., 2021; Vargas et al., 2021; Chen et al., 2022a; Peluchetti, 2023; Shi et al., 2023; Liu et al., 2023d). Two recent works (Liu et al., 2023a; Somnath et al., 2023) build the bridge in image translation and a biology task, while neither of them investigates the design space discussed in our work, which is of importance to sample quality and inference speed.

## 6 CONCLUSIONS

We present Bridge-TTS, a novel TTS method built on data-to-data process, enabling mel-spectrogram generation from a deterministic prior via Schrodinger bridge. Under our theoretically elaborated tractable, flexible SB framework, we exhaustively explore the design space of noise schedule, model parameterization, and stochastic/deterministic samplers. Experimental results on sample quality and sampling efficiency in TTS synthesis demonstrate the effectiveness of our approach, which significantly outperforms previous methods and becomes a new baseline on this task. We hope our work could open a new avenue for exploiting the board family of strong informative prior to further unleash the potential of generative models on a wide range of applications.

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

## CONTENTS

# A  PROOFS

## A.1  TRACTABLE SCHRODINGER BRIDGE BETWEEN GAUSSIAN-SMOOTHED PAIRED DATA

*Proof of Proposition 3.1.* First, we conduct a similar transformation to Liu et al. (2023a), which reverses the forward-backward SDE system of the SB in Eqn. (6) and absorb the intractable term $\widehat{\Psi}, \Psi$ into the boundary condition. On one hand, by inspecting the backward SDE (Eqn. (6b)) and its corresponding PDE (the second equation in Eqn. (7)), we can discover that if we regard $\widehat{\Psi}$ as a probability density function (up to a normalizing factor, which is canceled when we compute the score by operator $\nabla \log$), then the PDE of the backward SDE is a realization of the following forward SDE due to the Fokker-Plank equation (Song et al., 2021b):

$$\mathrm{d}\boldsymbol{x}_t = \boldsymbol{f}(\boldsymbol{x}_t, t)\mathrm{d}t + g(t)\mathrm{d}\boldsymbol{w}_t, \quad \boldsymbol{x}_0 \sim \widehat{\Psi}_0, \tag{21}$$

and its associated density of $\boldsymbol{x}_t$ is $\widehat{\Psi}_t$. When we assume $\boldsymbol{f}(\boldsymbol{x}_t, t) = f(t)\boldsymbol{x}_t$ as a linear drift, then Eqn. (21) becomes a *narrow-sense linear SDE*, whose conditional distribution $\widehat{\Psi}_{t|0}(\boldsymbol{x}_t|\boldsymbol{x}_0)$ is a tractable Gaussian, which we will prove as follows.

Specifically, Itô's formula (Itô, 1951) tells us that, for a general SDE with drift $\mu_t$ and diffusion $\sigma_t$:

$$\mathrm{d}x_t = \mu_t(x_t)\mathrm{d}t + \sigma_t(x_t)\mathrm{d}w_t \tag{22}$$

If $f(x, t)$ is a twice-differentiable function, then

$$\mathrm{d}f(x_t, t) = \left( \frac{\partial f}{\partial t}(x_t, t) + \mu_t(x_t)\frac{\partial f}{\partial x}(x_t, t) + \frac{\sigma_t^2(x_t)}{2}\frac{\partial^2 f}{\partial x^2}(x_t, t) \right) \mathrm{d}t + \sigma_t(x_t)\frac{\partial f}{\partial x}(x_t, t)\mathrm{d}w_t \tag{23}$$

Denote $\alpha_t = e^{\int_0^t f(\tau)\mathrm{d}\tau}$, if we choose $\boldsymbol{f}(\boldsymbol{x}, t) = \frac{\boldsymbol{x}}{\alpha_t}$, by Itô's formula we have

$$\mathrm{d}\left( \frac{\boldsymbol{x}_t}{\alpha_t} \right) = \frac{g(t)}{\alpha_t}\mathrm{d}\boldsymbol{w}_t \tag{24}$$

which clearly leads to the result

$$\frac{\boldsymbol{x}_t}{\alpha_t} - \frac{\boldsymbol{x}_0}{\alpha_0} \sim \mathcal{N}\left( \boldsymbol{0}, \int_0^t \frac{g^2(\tau)}{\alpha_\tau^2}\mathrm{d}\tau \boldsymbol{I} \right) \tag{25}$$

If we denote $\sigma_t^2 = \int_0^t \frac{g^2(\tau)}{\alpha_\tau^2}\mathrm{d}\tau$, finally we conclude that $\widehat{\Psi}_{t|0}(\boldsymbol{x}_t|\boldsymbol{x}_0) = \mathcal{N}(\alpha_t\boldsymbol{x}_0, \alpha_t^2\sigma_t^2\boldsymbol{I})$.

On the other hand, due to the symmetry of the SB, we can reverse the time $t$ by $s = 1-t$ and conduct similar derivations for $\Psi$, which finally leads to the result $\Psi_{t|1}(\boldsymbol{x}_t|\boldsymbol{x}_1) = \mathcal{N}(\bar{\alpha}_t\boldsymbol{x}_1, \alpha_t^2\bar{\sigma}_t^2\boldsymbol{I})$.

Since we have Gaussian boundary conditions:

$$p_{\mathrm{data}} = \widehat{\Psi}_0\Psi_0 = \mathcal{N}(\boldsymbol{x}_0, \epsilon^2\boldsymbol{I}), \quad p_{\mathrm{prior}} = \widehat{\Psi}_1\Psi_1 = \mathcal{N}(\boldsymbol{x}_1, \alpha_1^2\epsilon^2\boldsymbol{I}) \tag{26}$$

Due to the properties of Gaussian distribution, it is intuitive to assume that the marginal distributions $\widehat{\Psi}_0, \Psi_1$ are also Gaussian. We parameterize them with undetermined mean and variance as follows:

$$\widehat{\Psi}_0 = \mathcal{N}(\boldsymbol{a}, \sigma^2\boldsymbol{I}), \quad \Psi_1 = \mathcal{N}(\boldsymbol{b}, \alpha_1^2\sigma^2\boldsymbol{I}) \tag{27}$$

Since the conditional transitions $\widehat{\Psi}_{t|0}, \Psi_{t|1}$ are known Gaussian as we have derived, the marginals at any $t \in [0, 1]$ are also Gaussian (which can be seen as a simple linear Gaussian model):

$$\widehat{\Psi}_t = \mathcal{N}(\alpha_t\boldsymbol{a}, (\alpha_t^2\sigma^2 + \alpha_t^2\sigma_t^2)\boldsymbol{I}), \quad \Psi_t = \mathcal{N}(\bar{\alpha}_t\boldsymbol{b}, (\alpha_t^2\sigma^2 + \alpha_t^2\bar{\sigma}_t^2)\boldsymbol{I}) \tag{28}$$

Then we can solve the coefficients $\boldsymbol{a}, \boldsymbol{b}, \sigma$ by the boundary conditions. Note that $\bar{\sigma}_0^2 = \sigma_1^2, \bar{\alpha}_0 = \frac{1}{\alpha_1}$, and the product of two Gaussian probability density functions is given by

$$\mathcal{N}(\mu_1, \sigma_1^2)\mathcal{N}(\mu_2, \sigma_2^2) = \mathcal{N}\left( \frac{\sigma_2^2\mu_1 + \sigma_1^2\mu_2}{\sigma_1^2 + \sigma_2^2}, \frac{\sigma_1^2\sigma_2^2}{\sigma_1^2 + \sigma_2^2} \right) \tag{29}$$

We have

$$
\begin{cases}
\widehat{\Psi}_0\Psi_0 = \mathcal{N}(\boldsymbol{a}, \sigma^2\boldsymbol{I})\mathcal{N}(\bar{\alpha}_0\boldsymbol{b}, (\alpha_0^2\sigma^2 + \alpha_0^2\bar{\sigma}_0^2)\boldsymbol{I}) = \mathcal{N}(\boldsymbol{x}_0, \epsilon^2\boldsymbol{I}) \\
\widehat{\Psi}_1\Psi_1 = \mathcal{N}(\alpha_1\boldsymbol{a}, (\alpha_1^2\sigma^2 + \alpha_1^2\sigma_1^2)\boldsymbol{I})\mathcal{N}(\boldsymbol{b}, \alpha_1^2\sigma^2\boldsymbol{I}) = \mathcal{N}(\boldsymbol{x}_1, \alpha_1^2\epsilon^2\boldsymbol{I})
\end{cases}
\tag{30}
$$

$$
\Rightarrow
\begin{cases}
\dfrac{(\sigma^2 + \sigma_1^2)\boldsymbol{a} + \sigma^2\frac{\boldsymbol{b}}{\alpha_1}}{2\sigma^2 + \sigma_1^2} = \boldsymbol{x}_0 \\[2mm]
\dfrac{\alpha_1\sigma^2\boldsymbol{a} + (\sigma^2 + \sigma_1^2)\boldsymbol{b}}{2\sigma^2 + \sigma_1^2} = \boldsymbol{x}_1 \\[2mm]
\dfrac{\sigma^2(\sigma^2 + \sigma_1^2)}{2\sigma^2 + \sigma_1^2} = \epsilon^2
\end{cases}
\Rightarrow
\begin{cases}
\boldsymbol{a} = \boldsymbol{x}_0 + \dfrac{\sigma^2}{\sigma_1^2}\left(\boldsymbol{x}_0 - \dfrac{\boldsymbol{x}_1}{\alpha_1}\right) \\[2mm]
\boldsymbol{b} = \boldsymbol{x}_1 + \dfrac{\sigma^2}{\sigma_1^2}(\boldsymbol{x}_1 - \alpha_1\boldsymbol{x}_0) \\[2mm]
\sigma^2 = \epsilon^2 + \dfrac{\sqrt{\sigma_1^4 + 4\epsilon^4} - \sigma_1^2}{2}
\end{cases}
\tag{31}
$$

The proof is then completed by substituting these solved coefficients back into Eqn. (28). □

## A.2 Bridge Sampling

First of all, we would like to give some background information about exponential integrators (Calvo & Palencia, 2006; Hochbruck et al., 2009), which are widely used in recent works concerning fast sampling of diffusion ODE/SDEs (Lu et al., 2022a;b; Gonzalez et al., 2023). Suppose we have an SDE (or equivalently an ODE by setting $g(t) = 0$):

$$
\mathrm{d}\boldsymbol{x}_t = [a(t)\boldsymbol{x}_t + b(t)\boldsymbol{F}_\theta(\boldsymbol{x}_t, t)]\mathrm{d}t + g(t)\mathrm{d}\boldsymbol{w}_t
\tag{32}
$$

where $\boldsymbol{F}_\theta$ is the parameterized prediction function that we want to approximate with Taylor expansion. The usual way of representing its analytic solution $\boldsymbol{x}_t$ at time $t$ with respect to an initial condition $\boldsymbol{x}_s$ at time $s$ is

$$
\boldsymbol{x}_t = \boldsymbol{x}_s + \int_s^t [a(\tau)\boldsymbol{x}_\tau + b(\tau)\boldsymbol{F}_\theta(\boldsymbol{x}_\tau, \tau)]\mathrm{d}\tau + \int_s^t g(\tau)\mathrm{d}\boldsymbol{w}_\tau
\tag{33}
$$

By approximating the involved integrals in Eqn. (33), we can obtain direct discretizations of Eqn. (32) such as Euler's method. The key insight of exponential integrators is that, it is often better to utilize the "semi-linear" structure of Eqn. (32) and analytically cancel the linear term $a(t)\boldsymbol{x}_t$. This way, we can obtain solutions that only involve integrals of $\boldsymbol{F}_\theta$ and result in lower discretization errors. Specifically, by the "variation-of-constants" formula, the exact solution of Eqn. (32) can be alternatively given by

$$
\boldsymbol{x}_t = e^{\int_s^t a(\tau)\mathrm{d}\tau}\boldsymbol{x}_s + \int_s^t e^{\int_\tau^t a(r)\mathrm{d}r}b(\tau)\boldsymbol{F}_\theta(\boldsymbol{x}_\tau, \tau)\mathrm{d}\tau + \int_s^t e^{\int_\tau^t a(r)\mathrm{d}r}g(\tau)\mathrm{d}\boldsymbol{w}_\tau
\tag{34}
$$

or equivalently (assume $t < s$)

$$
\boldsymbol{x}_t = e^{\int_s^t a(\tau)\mathrm{d}\tau}\boldsymbol{x}_s + \int_s^t e^{\int_\tau^t a(r)\mathrm{d}r}b(\tau)\boldsymbol{F}_\theta(\boldsymbol{x}_\tau, \tau)\mathrm{d}\tau + \sqrt{-\int_s^t e^{2\int_\tau^t a(r)\mathrm{d}r}g^2(\tau)\mathrm{d}\tau}\boldsymbol{\epsilon}, \quad \boldsymbol{\epsilon} \sim \mathcal{N}(\mathbf{0}, \boldsymbol{I})
\tag{35}
$$

Then we prove Proposition 3.2 below.

*Proof of Proposition 3.2.* First, we consider the bridge SDE in Eqn. (14). By collecting the linear terms w.r.t. $\boldsymbol{x}_t$, the bridge SDE can be rewritten as

$$
\mathrm{d}\boldsymbol{x}_t = \left[\left(f(t) + \frac{g^2(t)}{\alpha_t^2\sigma_t^2}\right)\boldsymbol{x}_t - \frac{g^2(t)}{\alpha_t\sigma_t^2}\boldsymbol{x}_\theta(\boldsymbol{x}_t, t)\right]\mathrm{d}t + g(t)\mathrm{d}\boldsymbol{w}_t
\tag{36}
$$

By corresponding it to Eqn. (32), we have

$$
a(t) = f(t) + \frac{g^2(t)}{\alpha_t^2\sigma_t^2}, \quad b(t) = -\frac{g^2(t)}{\alpha_t\sigma_t^2}
\tag{37}
$$

The exponents in Eqn. (35) can be calculated as

$$
\int_s^t a(\tau)\mathrm{d}\tau = \int_s^t f(\tau)\mathrm{d}\tau + \int_s^t \frac{(\sigma_\tau^2)'}{\sigma_\tau^2}\mathrm{d}\tau = \int_s^t f(\tau)\mathrm{d}\tau + \log\frac{\sigma_t^2}{\sigma_s^2}
\tag{38}
$$

Thus
$$e^{\int_s^t a(\tau)\mathrm{d}\tau} = \frac{\alpha_t \sigma_t^2}{\alpha_s \sigma_s^2}, \quad e^{\int_\tau^t a(r)\mathrm{d}r} = \frac{\alpha_t \sigma_t^2}{\alpha_\tau \sigma_\tau^2} \tag{39}$$

Therefore, the exact solution in Eqn. (35) becomes
$$\boldsymbol{x}_t = \frac{\alpha_t \sigma_t^2}{\alpha_s \sigma_s^2}\boldsymbol{x}_s - \alpha_t \sigma_t^2 \int_s^t \frac{g^2(\tau)}{\alpha_\tau^2 \sigma_\tau^4}\boldsymbol{x}_\theta(\boldsymbol{x}_\tau,\tau)\mathrm{d}\tau + \alpha_t \sigma_t^2 \sqrt{-\int_s^t \frac{g^2(\tau)}{\alpha_\tau^2 \sigma_\tau^4}\mathrm{d}\tau}\,\boldsymbol{\epsilon}, \quad \boldsymbol{\epsilon} \sim \mathcal{N}(\mathbf{0},\boldsymbol{I}) \tag{40}$$

where
$$\int_s^t \frac{g^2(\tau)}{\alpha_\tau^2 \sigma_\tau^4}\mathrm{d}\tau = \int_s^t \frac{(\sigma_\tau^2)'}{\sigma_\tau^4}\mathrm{d}\tau = \frac{1}{\sigma_s^2} - \frac{1}{\sigma_t^2} \tag{41}$$

Substituting Eqn. (41) into Eqn. (40), we obtain the exact solution in Eqn. (17). If we take the first-order approximation (i.e., $\boldsymbol{x}_\theta(\boldsymbol{x}_\tau,\tau) \approx \boldsymbol{x}_\theta(\boldsymbol{x}_s,s)$ for $\tau \in [t,s]$), then we obtain the first-order transition rule in Eqn. (19).

Then we consider the bridge ODE in Eqn. (16). By collecting the linear terms w.r.t. $\boldsymbol{x}_t$, the bridge ODE can be rewritten as
$$\mathrm{d}\boldsymbol{x}_t = \left[\left(f(t) - \frac{g^2(t)}{2\alpha_t^2 \bar{\sigma}_t^2} + \frac{g^2(t)}{2\alpha_t^2 \sigma_t^2}\right)\boldsymbol{x}_t + \frac{g^2(t)\bar{\alpha}_t}{2\alpha_t^2 \bar{\sigma}_t^2}\boldsymbol{x}_1 - \frac{g^2(t)}{2\alpha_t \sigma_t^2}\boldsymbol{x}_\theta(\boldsymbol{x}_t,t)\right]\mathrm{d}t \tag{42}$$

By corresponding it to Eqn. (32), we have
$$a(t) = f(t) - \frac{g^2(t)}{2\alpha_t^2 \bar{\sigma}_t^2} + \frac{g^2(t)}{2\alpha_t^2 \sigma_t^2}, \quad b_1(t) = \frac{g^2(t)\bar{\alpha}_t}{2\alpha_t^2 \bar{\sigma}_t^2}, \quad b_2(t) = -\frac{g^2(t)}{2\alpha_t \sigma_t^2} \tag{43}$$

The exponents in Eqn. (35) can be calculated as
$$\begin{aligned}
\int_s^t a(\tau)\mathrm{d}\tau &= \int_s^t f(\tau)\mathrm{d}\tau - \int_s^t \frac{g^2(\tau)}{2\alpha_\tau^2 \bar{\sigma}_\tau^2}\mathrm{d}\tau + \int_s^t \frac{g^2(\tau)}{2\alpha_\tau^2 \sigma_\tau^2}\mathrm{d}\tau \\
&= \int_s^t f(\tau)\mathrm{d}\tau + \int_s^t \frac{(\bar{\sigma}_\tau^2)'}{2\bar{\sigma}_\tau^2}\mathrm{d}\tau + \int_s^t \frac{(\sigma_\tau^2)'}{2\sigma_\tau^2}\mathrm{d}\tau \\
&= \int_s^t f(\tau)\mathrm{d}\tau + \frac{1}{2}\log\frac{\bar{\sigma}_t^2}{\bar{\sigma}_s^2} + \frac{1}{2}\log\frac{\sigma_t^2}{\sigma_s^2}
\end{aligned} \tag{44}$$

Thus
$$e^{\int_s^t a(\tau)\mathrm{d}\tau} = \frac{\alpha_t \sigma_t \bar{\sigma}_t}{\alpha_s \sigma_s \bar{\sigma}_s}, \quad e^{\int_\tau^t a(r)\mathrm{d}r} = \frac{\alpha_t \sigma_t \bar{\sigma}_t}{\alpha_\tau \sigma_\tau \bar{\sigma}_\tau} \tag{45}$$

Therefore, the exact solution in Eqn. (35) becomes
$$\boldsymbol{x}_t = \frac{\alpha_t \sigma_t \bar{\sigma}_t}{\alpha_s \sigma_s \bar{\sigma}_s}\boldsymbol{x}_s + \frac{\bar{\alpha}_t \sigma_t \bar{\sigma}_t}{2}\int_s^t \frac{g^2(\tau)}{\alpha_\tau^2 \sigma_\tau \bar{\sigma}_\tau^3}\boldsymbol{x}_1 \mathrm{d}\tau - \frac{\alpha_t \sigma_t \bar{\sigma}_t}{2}\int_s^t \frac{g^2(\tau)}{\alpha_\tau^2 \sigma_\tau^3 \bar{\sigma}_\tau}\boldsymbol{x}_\theta(\boldsymbol{x}_\tau,\tau)\mathrm{d}\tau \tag{46}$$

Due the relation $\sigma_t^2 + \bar{\sigma}_t^2 = \sigma_1^2$, the integrals can be computed by the substitution $\theta_t = \arctan(\sigma_t/\bar{\sigma}_t)$
$$\begin{aligned}
\int_s^t \frac{g^2(\tau)}{\alpha_\tau^2 \sigma_\tau \bar{\sigma}_\tau^3}\mathrm{d}\tau &= \int_s^t \frac{(\sigma_\tau^2)'}{\sigma_\tau \bar{\sigma}_\tau^3}\mathrm{d}\tau \\
&= \int_{\theta_s}^{\theta_t} \frac{1}{\sigma_1^4 \sin\theta \cos^3\theta}\mathrm{d}(\sigma_1^2 \sin^2\theta) \\
&= \frac{2}{\sigma_1^2}\int_{\theta_s}^{\theta_t} \frac{1}{\cos^2\theta}\mathrm{d}\theta \\
&= \frac{2}{\sigma_1^2}(\tan\theta_t - \tan\theta_s) \\
&= \frac{2}{\sigma_1^2}\left(\frac{\sigma_t}{\bar{\sigma}_t} - \frac{\sigma_s}{\bar{\sigma}_s}\right)
\end{aligned} \tag{47}$$

and similarly
$$\int_s^t \frac{g^2(\tau)}{\alpha_\tau^2 \sigma_\tau^3 \bar{\sigma}_\tau}\mathrm{d}\tau = \frac{2}{\sigma_1^2}\left(\frac{\bar{\sigma}_s}{\sigma_s} - \frac{\bar{\sigma}_t}{\sigma_t}\right) \tag{48}$$

Substituting Eqn. (47) and Eqn. (48) into Eqn. (46), we obtain the exact solution in Eqn. (18). If we take the first-order approximation (i.e., $\boldsymbol{x}_\theta(\boldsymbol{x}_\tau,\tau) \approx \boldsymbol{x}_\theta(\boldsymbol{x}_s,s)$ for $\tau \in [t,s]$), then we obtain the first-order transition rule in Eqn. (20). $\qquad\square$

# B RELATIONSHIP WITH BROWNIAN BRIDGE, POSTERIOR SAMPLING AND DDIM

## B.1 SCHRODINGER BRIDGE PROBLEM AND BROWNIAN BRIDGE

For any path measure $\mu$ on $[0, 1]$, we have $\mu = \mu_{0,1}\mu_{|0,1}$, where $\mu_{0,1}$ denotes the joint distribution of $\mu_0, \mu_1$, and $\mu_{|0,1}$ denotes the conditional path measure on $(0, 1)$ given boundaries $\boldsymbol{x}_0, \boldsymbol{x}_1$. A high-level perspective is that, using the decomposition formula for KL divergence $D_{\mathrm{KL}}(p \parallel p^{\mathrm{ref}}) = D_{\mathrm{KL}}(p_{0,1} \parallel p_{0,1}^{\mathrm{ref}}) + D_{\mathrm{KL}}(p_{|0,1} \parallel p_{|0,1}^{\mathrm{ref}})$ (Léonard, 2014), the SB problem in Eqn. (5) can be reduced to the *static SB* problem (De Bortoli et al., 2021; Shi et al., 2023; Tong et al., 2023a;b):

$$\min_{p_{0,1} \in \mathcal{P}_2} D_{\mathrm{KL}}(p_{0,1} \parallel p_{0,1}^{\mathrm{ref}}), \quad s.t. \ p_0 = p_{\mathrm{data}}, p_1 = p_{\mathrm{prior}} \tag{49}$$

which is proved to be an *entropy-regularized optimal transport* problem when $p^{\mathrm{ref}}$ is defined by a scaled Brownian process $\mathrm{d}\boldsymbol{x}_t = \sigma \mathrm{d}\boldsymbol{w}_t$. We can draw similar conclusions for the more general case of reference SDE in Eqn. (1) with linear drift $\boldsymbol{f}(\boldsymbol{x}_t, t) = f(t)\boldsymbol{x}_t$. Specifically, the KL divergence between the joint distribution of boundaries is

$$\begin{aligned} D_{\mathrm{KL}}(p_{0,1} \parallel p_{0,1}^{\mathrm{ref}}) &= -\mathbb{E}_{p_{0,1}}[\log p_{0,1}^{\mathrm{ref}}] - \mathcal{H}(p_{0,1}) \\ &= -\mathbb{E}_{p_0}[\log p_0^{\mathrm{ref}}] - \mathbb{E}_{p_{0,1}}[\log p_{1|0}^{\mathrm{ref}}] - \mathcal{H}(p_{0,1}) \end{aligned} \tag{50}$$

where $\mathcal{H}(\cdot)$ is the entropy. As we have proved in Appendix A.1, $p_{t|0}^{\mathrm{ref}}(\boldsymbol{x}_t|\boldsymbol{x}_0) = \mathcal{N}(\alpha_t \boldsymbol{x}_0, \alpha_t^2 \sigma_t^2 \boldsymbol{I})$, thus

$$\log p_{1|0}^{\mathrm{ref}}(\boldsymbol{x}_1|\boldsymbol{x}_0) = -\frac{\|\boldsymbol{x}_1 - \alpha_1 \boldsymbol{x}_0\|_2^2}{2\alpha_1^2 \sigma_1^2} \tag{51}$$

Since $\mathbb{E}_{p_0}[\log p_0^{\mathrm{ref}}] = \mathbb{E}_{p_{\mathrm{data}}}[\log p_{\mathrm{data}}]$ is irrelevant to $p$, the static SB problem is equivalent to

$$\min_{p_{0,1} \in \mathcal{P}_2} \mathbb{E}_{p_{0,1}(\boldsymbol{x}_0, \boldsymbol{x}_1)}[\|\boldsymbol{x}_1 - \alpha_1 \boldsymbol{x}_0\|_2^2] - 2\alpha_1^2 \sigma_1^2 \mathcal{H}(p_{0,1}), \quad s.t. \ p_0 = p_{\mathrm{data}}, p_1 = p_{\mathrm{prior}} \tag{52}$$

Therefore, it is an entropy-regularized optimal transport problem when $\alpha_1 = 1$.

While the static SB problem is generally non-trivial, there exists application cases when we can skip it: when the coupling $p_{0,1}$ of $p_{\mathrm{data}}$ and $p_{\mathrm{prior}}$ is unique and has no room for further optimization. (1) When $p_{\mathrm{data}}$ is a Dirac delta distribution and $p_{\mathrm{prior}}$ is a usual distribution (Liu et al., 2023a). In this case, the SB is half tractable, and only the bridge SDE holds. (2) When paired data are considered, i.e., the coupling of $p_{\mathrm{data}}$ and $p_{\mathrm{prior}}$ is mixtures of dual Dirac delta distributions. In this case, however, $D_{\mathrm{KL}}(p_{0,1} \parallel p_{0,1}^{\mathrm{ref}}) = \infty$, and the SB problem will collapse. Still, we can ignore such singularity, so that the SB is fully tractable, and bridge ODE can be derived.

After the static SB problem is solved, we only need to minimize $D_{\mathrm{KL}}(p_{|0,1} \parallel p_{|0,1}^{\mathrm{ref}})$ in order to solve the original SB problem. In fact, since there is no constraints, such optimization directly leads to $p_{t|0,1} = p_{t|0,1}^{\mathrm{ref}}$ for $t \in (0, 1)$. When $p^{\mathrm{ref}}$ is defined by a scaled Brownian process $\mathrm{d}\boldsymbol{x}_t = \sigma \mathrm{d}\boldsymbol{w}_t$, $p_{t|0,1}^{\mathrm{ref}}$ is the common Brownian bridge (Qiu et al., 2023; Tong et al., 2023a;b). When $p^{\mathrm{ref}}$ is defined by the narrow-sense linear SDE $\mathrm{d}\boldsymbol{x}_t = f(t)\boldsymbol{x}_t \mathrm{d}t + g(t)\mathrm{d}\boldsymbol{w}_t$ which we considered, $p_{t|0,1}^{\mathrm{ref}}$ can be seen as the generalized Brownian bridge with linear drift and time-varying volatility, and we can derive its formula as follows.

Similar to the derivations in Appendix A.1, the transition probability from time $s$ to time $t$ $(s < t)$ following the reference SDE $\mathrm{d}\boldsymbol{x}_t = f(t)\boldsymbol{x}_t \mathrm{d}t + g(t)\mathrm{d}\boldsymbol{w}_t$ is

$$p_{t|s}^{\mathrm{ref}}(\boldsymbol{x}_t|\boldsymbol{x}_s) = \mathcal{N}(\boldsymbol{x}_t; \alpha_{t|s}\boldsymbol{x}_s, \alpha_{t|s}^2 \sigma_{t|s}^2 \boldsymbol{I}) \tag{53}$$

where $\alpha_{t|s}, \sigma_{t|s}$ are the corresponding coefficients to $\alpha_t, \sigma_t$, while modifying the lower limit of integrals from 0 to $s$:

$$\alpha_{t|s} = e^{\int_s^t f(\tau)\mathrm{d}\tau}, \quad \sigma_{t|s}^2 = \int_s^t \frac{g^2(\tau)}{\alpha_{\tau|s}^2}\mathrm{d}\tau \tag{54}$$

We can easily identify that $\alpha_{t|s}, \sigma_{t|s}$ are related to $\alpha_t, \sigma_t$ by

$$\alpha_{t|s} = \frac{\alpha_t}{\alpha_s}, \quad \sigma_{t|s}^2 = \alpha_s^2(\sigma_t^2 - \sigma_s^2) \tag{55}$$

Therefore

$$p_{t|s}^{\text{ref}}(\boldsymbol{x}_t|\boldsymbol{x}_s) = \mathcal{N}\left(\boldsymbol{x}_t; \frac{\alpha_t}{\alpha_s}\boldsymbol{x}_s, \alpha_t^2(\sigma_t^2 - \sigma_s^2)\boldsymbol{I}\right) \tag{56}$$

Due to the Markov property of the SDE, we can compute $p_{t|0,1}^{\text{ref}}$ as

$$
\begin{aligned}
p_{t|0,1}^{\text{ref}}(\boldsymbol{x}_t|\boldsymbol{x}_0, \boldsymbol{x}_1) &= \frac{p_{t,1|0}^{\text{ref}}(\boldsymbol{x}_t, \boldsymbol{x}_1|\boldsymbol{x}_0)}{p_{1|0}^{\text{ref}}(\boldsymbol{x}_1|\boldsymbol{x}_0)} \\
&= \frac{p_{t|0}^{\text{ref}}(\boldsymbol{x}_t|\boldsymbol{x}_0)p_{1|t}^{\text{ref}}(\boldsymbol{x}_1|\boldsymbol{x}_t)}{p_{1|0}^{\text{ref}}(\boldsymbol{x}_1|\boldsymbol{x}_0)} \\
&\propto \frac{\exp\left(-\frac{\|\boldsymbol{x}_t - \alpha_t\boldsymbol{x}_0\|_2^2}{2\alpha_t^2\sigma_t^2}\right)\exp\left(-\frac{\|\boldsymbol{x}_1 - \frac{\alpha_1}{\alpha_t}\boldsymbol{x}_t\|_2^2}{2\alpha_1^2(\sigma_1^2 - \sigma_t^2)}\right)}{\exp\left(-\frac{\|\boldsymbol{x}_1 - \alpha_1\boldsymbol{x}_0\|_2^2}{2\alpha_1^2\sigma_1^2}\right)} \\
&\propto \exp\left(-\frac{\|\boldsymbol{x}_t - \alpha_t\boldsymbol{x}_0\|_2^2}{2\alpha_t^2\sigma_t^2} - \frac{\|\boldsymbol{x}_t - \bar{\alpha}_t\boldsymbol{x}_1\|_2^2}{2\alpha_t^2\bar{\sigma}_t^2}\right) \\
&\propto \exp\left(-\frac{\|\boldsymbol{x}_t - \frac{\alpha_t\bar{\sigma}_t^2\boldsymbol{x}_0 + \bar{\alpha}_t\sigma_t^2\boldsymbol{x}_1}{\sigma_1^2}\|_2^2}{2\frac{\alpha_t^2\sigma_t^2\bar{\sigma}_t^2}{\sigma_1^2}}\right)
\end{aligned}
\tag{57}
$$

Therefore, $p_{t|0,1}^{\text{ref}} = \mathcal{N}\left(\frac{\alpha_t\bar{\sigma}_t^2\boldsymbol{x}_0 + \bar{\alpha}_t\sigma_t^2\boldsymbol{x}_1}{\sigma_1^2}, \frac{\alpha_t^2\bar{\sigma}_t^2\sigma_t^2}{\sigma_1^2}\boldsymbol{I}\right)$, which equals the SB marginal in Eqn. (12).

### B.2 POSTERIOR SAMPLING ON A BROWNIAN BRIDGE AND DDIM

**Posterior Sampling and Bridge SDE** Liu et al. (2023a) proposes a method called *posterior sampling* to sample from bridge: when $p^{\text{ref}}$ is defined by $\mathrm{d}\boldsymbol{x}_t = \sqrt{\beta_t}\mathrm{d}\boldsymbol{w}_t$, we can sample $\boldsymbol{x}_{N-1}, \ldots, \boldsymbol{x}_{n+1}, \boldsymbol{x}_n, \ldots, \boldsymbol{x}_0$ at timesteps $t_{N-1}, \ldots, t_{n+1}, t_n, \ldots, t_0$ sequentially, where at each step the sample is generated from the DDPM posterior (Ho et al., 2020):

$$p(\boldsymbol{x}_n|\boldsymbol{x}_0, \boldsymbol{x}_{n+1}) = \mathcal{N}\left(\boldsymbol{x}_n; \frac{\alpha_n^2}{\alpha_n^2 + \sigma_n^2}\boldsymbol{x}_0 + \frac{\sigma_n^2}{\alpha_n^2 + \sigma_n^2}\boldsymbol{x}_{n+1}, \frac{\sigma_n^2\alpha_n^2}{\alpha_n^2 + \sigma_n^2}\boldsymbol{I}\right), \tag{58}$$

where $\alpha_n^2 = \int_{t_n}^{t_{n+1}} \beta(\tau)\mathrm{d}\tau$ is the accumulated noise between two timesteps $(t_n, t_{n+1})$, $\sigma_n^2 = \int_0^{t_n} \beta(\tau)\mathrm{d}\tau$, and $\boldsymbol{x}_0$ is predicted by the network.

While they only consider $f(t) = 0$ and prove the case for discrete timesteps by onerous mathematical induction, such posterior is essentially a "shortened" Brownian bridge. Suppose we already draw a sample $\boldsymbol{x}_s \sim p_{s|0,1}^{\text{ref}}$, then the sample at time $t < s$ can be drawn from $p_{t|0,1,s}^{\text{ref}}$, which equals $p_{t|0,s}^{\text{ref}}$ due to the Markov property of the SDE. Similar to the derivation in Eqn. (57), such shortened Brownian bridge is

$$p_{t|0,s}^{\text{ref}}(\boldsymbol{x}_t|\boldsymbol{x}_0, \boldsymbol{x}_s) = \mathcal{N}\left(\boldsymbol{x}_t; \frac{\alpha_t(\sigma_s^2 - \sigma_t^2)\boldsymbol{x}_0 + \frac{\alpha_t}{\alpha_s}\sigma_t^2\boldsymbol{x}_s}{\sigma_s^2}, \frac{\alpha_t^2\sigma_t^2(\sigma_s^2 - \sigma_t^2)}{\sigma_s^2}\boldsymbol{I}\right) \tag{59}$$

which is exactly the same as the first-order discretization of bridge SDE in Eqn. (19) when $\boldsymbol{x}_0$ is predicted by the network $\boldsymbol{x}_\theta(\boldsymbol{x}_s, s)$.

**DDIM and Bridge ODE** DDIM (Song et al., 2021a) is a sampling method for diffusion models, whose deterministic case is later proved to be the first-order discretization of certain solution forms of the diffusion ODE (Lu et al., 2022a;b). Under our notations of $\alpha_t, \sigma_t^2$, the update rule of DDIM is (Lu et al., 2022b)

$$\boldsymbol{x}_t = \frac{\alpha_t\sigma_t}{\alpha_s\sigma_s}\boldsymbol{x}_s + \alpha_t\left(1 - \frac{\sigma_t^2}{\sigma_s^2}\right)\boldsymbol{x}_\theta(\boldsymbol{x}_s, s) \tag{60}$$

In the limit of $\frac{\sigma_s}{\sigma_1}, \frac{\sigma_t}{\sigma_1} \to 0$, we have $\frac{\bar{\sigma}_s}{\sigma_1}, \frac{\bar{\sigma}_t}{\sigma_1} \to 1$. Therefore, $\frac{\bar{\sigma}_t}{\bar{\sigma}_s} \to 1$, and we can discover that Eqn. (20) reduces to Eqn. (60).

**Corollary B.1** (1-step First-Order Bridge SDE/ODE Sampler Recovers Direct Data Prediction).
*When $s = 1$ and $t = 0$, the first-order discretization of bridge SDE/ODE is*

$$\boldsymbol{x}_0 = \boldsymbol{x}_\theta(\boldsymbol{x}_1, 1) \tag{61}$$

## C  HIGH-ORDER SAMPLERS

We can develop high-order samplers by approximating $\boldsymbol{x}_\theta(\boldsymbol{x}_\tau, \tau), \tau \in [t, s]$ with high-order Taylor expansions. Specifically, we take the second-order case of the bridge SDE as an example. For the integral $\int_s^t \frac{g^2(\tau)}{\alpha_\tau^2 \sigma_\tau^4} \boldsymbol{x}_\theta(\boldsymbol{x}_\tau, \tau) \mathrm{d}\tau$ in Eqn. (17), we can use the change-of-variable $\lambda_t = -\frac{1}{\sigma_t^2}$. Since $(\lambda_t)' = \frac{g^2(t)}{\alpha_t^2 \sigma_t^4}$, the integral becomes

$$\begin{aligned}
\int_s^t \frac{g^2(\tau)}{\alpha_\tau^2 \sigma_\tau^4} \boldsymbol{x}_\theta(\boldsymbol{x}_\tau, \tau) \mathrm{d}\tau &= \int_{\lambda_s}^{\lambda_t} \boldsymbol{x}_\theta(\boldsymbol{x}_{\tau_\lambda}, \tau_\lambda) \mathrm{d}\lambda \\
&\approx \int_{\lambda_s}^{\lambda_t} \boldsymbol{x}_\theta(\boldsymbol{x}_s, s) + (\lambda - \lambda_s) \boldsymbol{x}_\theta^{(1)}(\boldsymbol{x}_s, s) \mathrm{d}\lambda \\
&= (\lambda_t - \lambda_s) \boldsymbol{x}_\theta(\boldsymbol{x}_s, s) + \frac{(\lambda_t - \lambda_s)^2}{2} \boldsymbol{x}_\theta^{(1)}(\boldsymbol{x}_s, s)
\end{aligned} \tag{62}$$

where $\tau_\lambda$ is the inverse mapping of $\lambda_\tau$, $\boldsymbol{x}_\theta^{(1)}$ is the first-order derivative of $\boldsymbol{x}_\theta$ w.r.t $\lambda$, and we have used the second-order Taylor expansion $\boldsymbol{x}_\theta(\boldsymbol{x}_{\tau_\lambda}, \tau_\lambda) \approx \boldsymbol{x}_\theta(\boldsymbol{x}_s, s) + (\lambda - \lambda_s) \boldsymbol{x}_\theta^{(1)}(\boldsymbol{x}_s, s)$. $\boldsymbol{x}_\theta^{(1)}$ can be estimated by finite difference, and a simple treatment is the predictor-corrector method. We first compute $\hat{\boldsymbol{x}}_t$ by the first-order update rule in Eqn. (19), which is used to estimate $\boldsymbol{x}_\theta^{(1)}(\boldsymbol{x}_s, s)$: $\boldsymbol{x}_\theta^{(1)}(\boldsymbol{x}_s, s) \approx \frac{\boldsymbol{x}_\theta(\hat{\boldsymbol{x}}_t, t) - \boldsymbol{x}_\theta(\boldsymbol{x}_s, s)}{\lambda_t - \lambda_s}$. Substituting it into Eqn. (62), we have $\int_s^t \frac{g^2(\tau)}{\alpha_\tau^2 \sigma_\tau^4} \boldsymbol{x}_\theta(\boldsymbol{x}_\tau, \tau) \mathrm{d}\tau \approx (\lambda_t - \lambda_s) \frac{\boldsymbol{x}_\theta(\boldsymbol{x}_s, s) + \boldsymbol{x}_\theta(\hat{\boldsymbol{x}}_t, t)}{2}$ which literally can be seen as replacing $\boldsymbol{x}_\theta(\boldsymbol{x}_s, s)$ in Eqn. (19) with $\frac{\boldsymbol{x}_\theta(\boldsymbol{x}_s, s) + \boldsymbol{x}_\theta(\hat{\boldsymbol{x}}_t, t)}{2}$. Similar derivations can be done for the bridge ODE. We summarize the second-order samplers in Algorithm 1 and Algorithm 2.

---

**Algorithm 1** Second-order sampler for the bridge SDE

---

**Input:** Number of function evaluations (NFE) $2N$, timesteps $1 = t_N > t_{N-1} > \cdots > t_n > t_{n-1} > \cdots > t_0 = 0$, initial condition $\boldsymbol{x}_1$

1: **for** $n = N$ **to** 1 **do**
2: $\quad s \leftarrow t_n$
3: $\quad t \leftarrow t_{n-1}$
4: $\quad$ Prediction: $\hat{\boldsymbol{x}}_t \leftarrow \frac{\alpha_t \sigma_t^2}{\alpha_s \sigma_s^2} \boldsymbol{x}_s + \alpha_t \left(1 - \frac{\sigma_t^2}{\sigma_s^2}\right) \boldsymbol{x}_\theta(\boldsymbol{x}_s, s) + \alpha_t \sigma_t \sqrt{1 - \frac{\sigma_t^2}{\sigma_s^2}} \boldsymbol{\epsilon}, \quad \boldsymbol{\epsilon} \sim \mathcal{N}(\boldsymbol{0}, \boldsymbol{I})$
5: $\quad$ Correction: $\boldsymbol{x}_t \leftarrow \frac{\alpha_t \sigma_t^2}{\alpha_s \sigma_s^2} \boldsymbol{x}_s + \alpha_t \left(1 - \frac{\sigma_t^2}{\sigma_s^2}\right) \frac{\boldsymbol{x}_\theta(\boldsymbol{x}_s, s) + \boldsymbol{x}_\theta(\hat{\boldsymbol{x}}_t, t)}{2} + \alpha_t \sigma_t \sqrt{1 - \frac{\sigma_t^2}{\sigma_s^2}} \boldsymbol{\epsilon}, \quad \boldsymbol{\epsilon} \sim \mathcal{N}(\boldsymbol{0}, \boldsymbol{I})$
6: **end for**

**Output:** $\boldsymbol{x}_0$

---

**Algorithm 2** Second-order sampler for the bridge ODE

---

**Input:** Number of function evaluations (NFE) $2N$, timesteps $1 = t_N > t_{N-1} > \cdots > t_n > t_{n-1} > \cdots > t_0 = 0$, initial condition $\boldsymbol{x}_1$

1: **for** $n = N$ **to** 1 **do**
2: $\quad s \leftarrow t_n$
3: $\quad t \leftarrow t_{n-1}$
4: $\quad$ Prediction: $\hat{\boldsymbol{x}}_t \leftarrow \frac{\alpha_t \sigma_t \bar{\sigma}_t}{\alpha_s \sigma_s \bar{\sigma}_s} \boldsymbol{x}_s + \frac{\alpha_t}{\sigma_1^2} \left[\left(\bar{\sigma}_t^2 - \frac{\bar{\sigma}_s \sigma_t \bar{\sigma}_t}{\sigma_s}\right) \boldsymbol{x}_\theta(\boldsymbol{x}_s, s) + \left(\sigma_t^2 - \frac{\sigma_s \sigma_t \bar{\sigma}_t}{\bar{\sigma}_s}\right) \frac{\boldsymbol{x}_1}{\alpha_1}\right]$
5: $\quad$ Correction: $\boldsymbol{x}_t \leftarrow \frac{\alpha_t \sigma_t \bar{\sigma}_t}{\alpha_s \sigma_s \bar{\sigma}_s} \boldsymbol{x}_s + \frac{\alpha_t}{\sigma_1^2} \left[\left(\bar{\sigma}_t^2 - \frac{\bar{\sigma}_s \sigma_t \bar{\sigma}_t}{\sigma_s}\right) \frac{\boldsymbol{x}_\theta(\boldsymbol{x}_s, s) + \boldsymbol{x}_\theta(\hat{\boldsymbol{x}}_t, t)}{2} + \left(\sigma_t^2 - \frac{\sigma_s \sigma_t \bar{\sigma}_t}{\bar{\sigma}_s}\right) \frac{\boldsymbol{x}_1}{\alpha_1}\right]$
6: **end for**

**Output:** $\boldsymbol{x}_0$

---

## D  MODEL PARAMETERIZATION

Apart from $x_0$ predictor $x_\theta$ presented in Section 3.2, we can consider other parameterizations:

- Noise predictor $\epsilon_\theta^{\widehat{\Psi}}$ corresponding to $\nabla \log \widehat{\Psi}_t = -\frac{x_t - \alpha_t x_0}{\alpha_t^2 \sigma_t^2}$ that used in I2SB (Liu et al., 2023a). The prediction target of $\epsilon_\theta^{\widehat{\Psi}}$ is:

$$\epsilon_\theta^{\widehat{\Psi}} \to \frac{x_t - \alpha_t x_0}{\alpha_t \sigma_t} \tag{63}$$

- Noise predictor $\epsilon_\theta^{\text{SB}}$ corresponding to the score $\nabla \log p_t$ of the SB. Since $\nabla \log p_t(x_t) = -\frac{x_t - \frac{\alpha_t \bar{\sigma}_t^2 x_0 + \bar{\alpha}_t \sigma_t^2 x_1}{\sigma_1^2}}{\frac{\alpha_t^2 \bar{\sigma}_t^2 \sigma_t^2}{\sigma_1^2}}$, the prediction target of $\epsilon_\theta^{\text{SB}}$ is

$$\epsilon_\theta^{\text{SB}} \to \frac{x_t - \frac{\alpha_t \bar{\sigma}_t^2 x_0 + \bar{\alpha}_t \sigma_t^2 x_1}{\sigma_1^2}}{\frac{\alpha_t \bar{\sigma}_t \sigma_t}{\sigma_1}} \tag{64}$$

- Velocity predictor $v_\theta$ arising from flow matching techniques (Lipman et al., 2023; Tong et al., 2023b;a), which aims to directly predict the drift of the PF-ODE:

$$v_\theta \to f(t)x_t - \frac{1}{2}g^2(t)\frac{x_t - \bar{\alpha}_t x_1}{\alpha_t^2 \bar{\sigma}_t^2} + \frac{1}{2}g^2(t)\frac{x_t - \alpha_t x_0}{\alpha_t^2 \sigma_t^2} \tag{65}$$

Empirically, across all parameterizations, we observe that the $x_0$ predictor and the noise predictor $\epsilon_\theta^{\widehat{\Psi}}$ work well in the TTS task and Table 5 shows that the $x_0$ predictor is generally better in sample quality. Hence, we adopt the $x_0$ predictor as the default training setup for Bridge-TTS. For the $\epsilon_\theta^{\text{SB}}$ predictor and $v_\theta$ predictor, we find that they lead to poor performance on the TTS task. We can intuitively explain this phenomenon by taking a simple case $f(t) = 0, g(t) = \sigma$. In this case, we have $x_t = (1-t)x_0 + tx_1 + \sigma\sqrt{t(1-t)}\epsilon, \epsilon \sim \mathcal{N}(0, I)$, and the prediction targets are

$$\begin{aligned}
x_\theta &\to x_0 \\
\epsilon_\theta^{\widehat{\Psi}} &\to \frac{x_t - x_0}{\sigma\sqrt{t}} = \sqrt{t}(x_1 - x_0) + \sigma\sqrt{1-t}\epsilon \\
\epsilon_\theta^{\text{SB}} &\to \frac{x_t - (1-t)x_0 - tx_1}{\sigma\sqrt{t(1-t)}} = \epsilon \\
\sqrt{t(1-t)}v_\theta &\to \frac{(1-2t)x_t - (1-t)x_0 + tx_1}{2\sqrt{t(1-t)}} = \sqrt{t(1-t)}(x_1 - x_0) + \sigma\frac{1-2t}{2}\epsilon
\end{aligned} \tag{66}$$

Therefore, $\epsilon_\theta^{\text{SB}}$ and $v_\theta$ both predict $\epsilon$ when $t \to 1$, while $x_\theta$ and $\epsilon_\theta^{\widehat{\Psi}}$ tends to predict $x_0, x_1$-related terms in such scenario. We can conclude that the former way of prediction is harmful on TTS task.

Table 5: CMOS comparison of different parameterizations of Bridge-TTS.

| Method | NFE=4 | NFE=1000 |
|---|---|---|
| Bridge-TTS (gmax + $x_0$ predictor) | 0 | 0 |
| Bridge-TTS (gmax + $\epsilon_\theta^{\widehat{\Psi}}$ predictor) | - 0.15 | - 0.12 |

# E  FORWARD PROCESS

In this section, we display the stochastic trajectory of the Bridge-SDE in Eqn. (14) and compare it with the diffusion counterpart in Eqn. (4). In general, the marginal distribution of these SDEs shares the form $p_t = \mathcal{N}(\boldsymbol{x}_t; \boldsymbol{w}_t \boldsymbol{x}_0 + \bar{\boldsymbol{w}}_t \boldsymbol{x}_1, \tilde{\sigma}_t^2 \boldsymbol{I})$. In Figure 3, we show the scaling factors $\boldsymbol{w}_t$ and $\bar{\boldsymbol{w}}_t$ for $\boldsymbol{x}_0$ and $\boldsymbol{x}_1$ and the variance $\tilde{\sigma}_t^2$ at time $t$. As described in Section 4.4, the Bridge-gmax and Bridge-VP have an asymmetric pattern of marginal variance that uses more steps to denoise towards the ground truth $\boldsymbol{x}_0$, while the constant $g(t)$ schedule specifies the same noise-additive and denoising steps. As a comparison, the diffusion-based model only performs denoising steps.

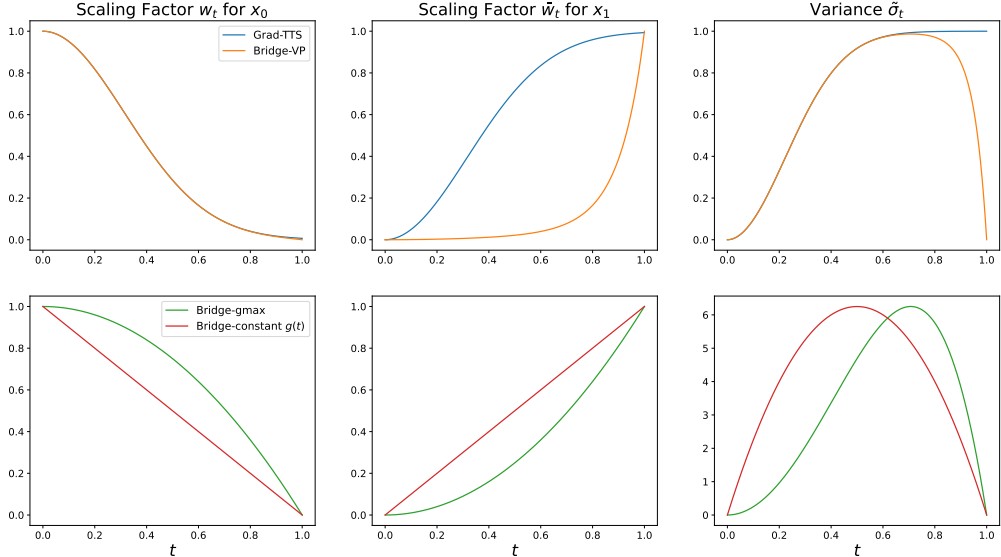

Figure 3: The scaling factor and variance in Grad-TTS and Bridge-TTS.

# F  BASELINE MODELS

Apart from ground-truth recording and the sample synthesized by vocoder from ground-truth mel-spectrogram, we take seven diffusion-based TTS systems, one end-to-end TTS system, and one transformer-based TTS model as our baseline models. We follow their official implementation or the settings reported in their publications to produce the results. We introduce each of our baseline models below:

**1. FastSpeech 2** (Ren et al., 2021) (ICLR 2021) is one of the most popular non-autoregressive TTS models, and widely used as the baseline in previous diffusion-based TTS systems (Chen et al., 2022c; Liu et al., 2022a; Ye et al., 2023). Following its original setting, we train the model with a batch size of 48 sentences and 160k training steps until convergence by using 8 NVIDIA V100 GPU.

**2. VITS** (Kim et al., 2021) (ICML 2021) provides a strong baseline of end-to-end TTS systems and is widely taken as a baseline in TTS systems for sample quality comparison. Different from other baseline models using pre-trained vocoder to generate waveform, VITS directly synthesizes waveform from text input. In training and testing, we follow their open-source implementation[4].

**3. DiffSinger** (Liu et al., 2022a) (AAAI 2022) is a TTS model developed for TTS synthesis and text-to-singing synthesis. It is built on denoising diffusion probabilistic models (Ho et al., 2020), using standard Gaussian noise $\mathcal{N}(\mathbf{0}, \mathbf{I})$ in the diffusion process. Moreover, an auxiliary model is trained to enable its shallow reverse process, *i.e.*, reducing the distance between prior distribution and data distribution. We follow their open-source implementation[5], which contains a warm-up stage for auxiliary model training and a main stage for diffusion model training.

**4. DiffGAN-TTS** (Liu et al., 2022b)[6] develops expressive generator and time-dependent discriminator to learn the non-Gaussian denoising distribution (Xiao et al., 2022) in few-step sampling process of diffusion models. Following their publication, we train DiffGAN-TTS with time steps $T = 4$. For both the generator and the discriminator, we use the Adam optimizer, with $\beta_1 = 0.5$ and $\beta_2 = 0.9$. Models are trained using a single NVIDIA V100 GPU. We set the batch size as 32, and train models for 400k steps until loss converges.

**5. ProDiff** (Huang et al., 2022) (ACM Multimedia 2022) is a fast TTS model using progressive distillation (Salimans & Ho, 2022). The standard Gaussian noise $\mathcal{N}(\mathbf{0}, \mathbf{I})$ is used in the diffusion process and taken as the prior distribution. We use their 2-step diffusion-based student model, which is distilled from a 4-step diffusion-based teacher model ($\boldsymbol{x}_0$ prediction). We follow their open-source implementation[7].

**6. Grad-TTS** (Popov et al., 2021)[8] (ICML 2021) is a widely used baseline in diffusion models (Huang et al., 2022; Chen et al., 2022c; 2023; Ye et al., 2023) and conditional flow matching (Mehta et al., 2023; Guo et al., 2023) based TTS systems. It is established on SGMs, providing a strong baseline of generation quality. Moreover, it realizes fast sampling with the improved prior distribution $\mathcal{N}(\boldsymbol{\mu}, \boldsymbol{I})$ and the temperature parameter $\tau = 1.5$ in inference. Following its original setting and publicly available implementation, we train the model with a batch size of 16 and 1.7 million steps on 1 NVIDIA 2080 GPU. The Adam optimizer is used and the learning rate is set to a constant, 0.0001.

**7. FastGrad-TTS** (Vovk et al., 2022) (INTERSPEECH 2022) equips pre-trained Grad-TTS (Popov et al., 2021) with the first-order SDE sampler proposed by (Popov et al., 2022). The Maximum Likelihood solver reduces the mismatch between the reverse and the forward process. In comparison with the first-order Euler scheme, this solver has shown improved quality in both voice conversion and TTS synthesis. We implement it for the pre-trained Grad-TTS model with the Equation (6)-(9) in its publication.

**8. ResGrad** (Chen et al., 2022c) is a diffusion-based post-processing module to improve the TTS sample quality, where the residual information of a pre-trained FastSpeech 2 (Ren et al., 2021) model

---

[4] https://github.com/jaywalnut310/vits
[5] https://github.com/MoonInTheRiver/DiffSinger
[6] https://github.com/keonlee9420/DiffGAN-TTS
[7] https://github.com/Rongjiehuang/ProDiff
[8] https://github.com/huawei-noah/Speech-Backbones/tree/main/Grad-TTS

is generated by a diffusion model. The standard Gaussian noise $\mathcal{N}(\mathbf{0}, \mathbf{I})$ is used in the diffusion process and taken as prior. We invite the authors to generate some test samples for us.

**9. CoMoSpeech** (Ye et al., 2023)[9] (ACM Multimedia 2023) is a recent fast sampling method in TTS and text-to-singing synthesis, achieving one-step generation with the distillation technique in consistency models (Song et al., 2023). As Grad-TTS is employed as its TTS backbone, the model uses $\mathcal{N}(\boldsymbol{\mu}, \mathbf{I})$ as prior distribution and is trained for 1.7 million iterations on a single NVIDIA A100 GPU with a batch size of 16. The Adam optimizer is adopted with a learning rate 0.0001.

## G   ADDITIONAL RESULTS

### G.1   PREFERENCE TEST

Apart from using the MOS test to evaluate sample quality, we conducted a blind preference test when NFE=1000 and NFE=2, in order to demonstrate our superior generation quality and efficient sampling process, respectively. In each test, we generated 100 identical samples with two different models from the test set LJ001 and LJ002, and invited 11 judges to compare their overall subjective quality. The judge gives a preference when he thinks a model is better than the other, and an identical result when he thinks it is hard to tell the difference or the models have similar overall quality. In both preference tests, the settings of noise schedule, model parameterization and sampling process in Bridge-TTS are $g_{max} = 50$, $x_0$ prediction, and first-order SDE sampler with $\tau_b = 2$, respectively.

In the case of NFE=1000, as shown in Figure 4 (a), when Bridge-TTS-1000 is compared with our diffusion counterpart Grad-TTS-1000 (Popov et al., 2021) (temperature $\tau_d = 1.5$), 8 of the 11 invited judges vote for Bridge-TTS-1000, and 3 of them think the overall quality is similar. In our blind test, none of the 11 judges preferred Grad-TTS-1000 to Bridge-TTS-1000.

In the case of NFE=2, as shown in Figure 4 (b), when Bridge-TTS-2 is compared with state-of-the-art fast sampling method in diffusion-based TTS systems, CoMoSpeech (1-step generation) (Ye et al., 2023), 9 of the 11 invited judges vote for Bridge-TTS-2, and 2 of the judges vote for CoMoSpeech-1. Although Bridge-TTS employs 2 sampling steps while CoMoSpeech-1 only uses 1, the RTF of both methods have been very small (0.007 for CoMoSpeech-1 vs 0.009 for Bridge-TTS-2), and Bridge-TTS does not require any distillation process. According to our collected feedback, 9 judges think the overall quality (e.g., quality, naturalness, and accuracy) of Bridge-TTS is distinctively better.

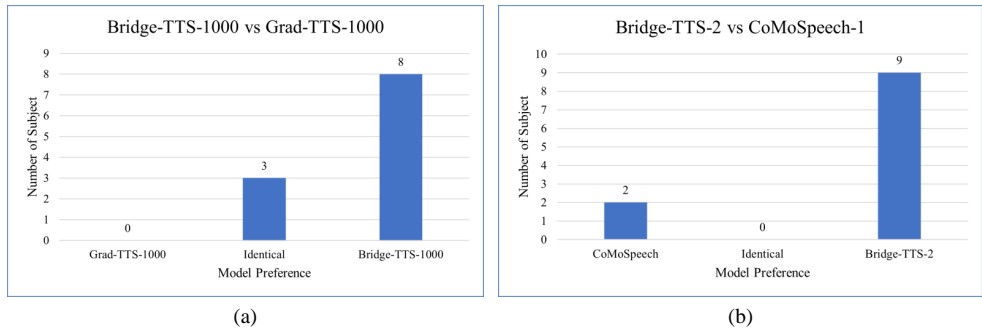

(a)                                                                    (b)

Figure 4: The preference test between Bridge-TTS and diffusion-based TTS systems.

### G.2   ADDITIONAL SAMPLES

With the pre-trained HiFi-GAN (Kong et al., 2020) vocoder, we show the 80-band mel-spectrogram of several synthesized test samples of baseline models and our Bridge-TTS (schedule $g_{max} = 50$, $x_0$ prediction, and temperature $\tau_b = 2$) below. The mel-spectrogram of ground-truth recording is shown for comparison.

---

[9]https://github.com/zhenye234/CoMoSpeech

**1000-step generation** As exhibited in Figure 5 and Figure 6, when NFE=1000, our method generates higher-quality speech than Grad-TTS (temperature $\tau_d = 1.5$) built on data-to-noise process, demonstrating the advantage of our proposed data-to-data process over data-to-noise process in TTS.

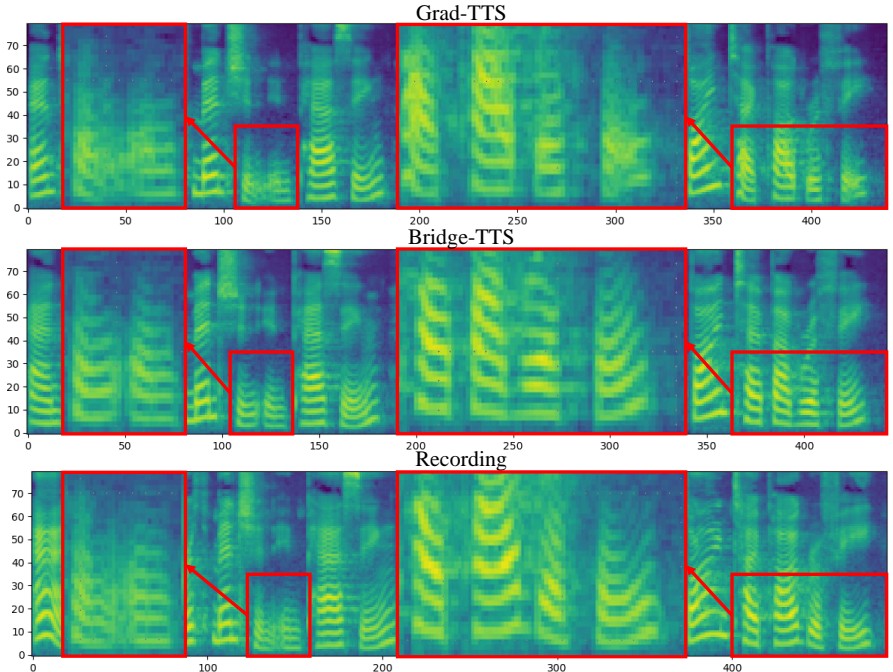

Figure 5: The mel-spectrogram of synthesized (NFE=1000) and ground-truth sample LJ001-0006.

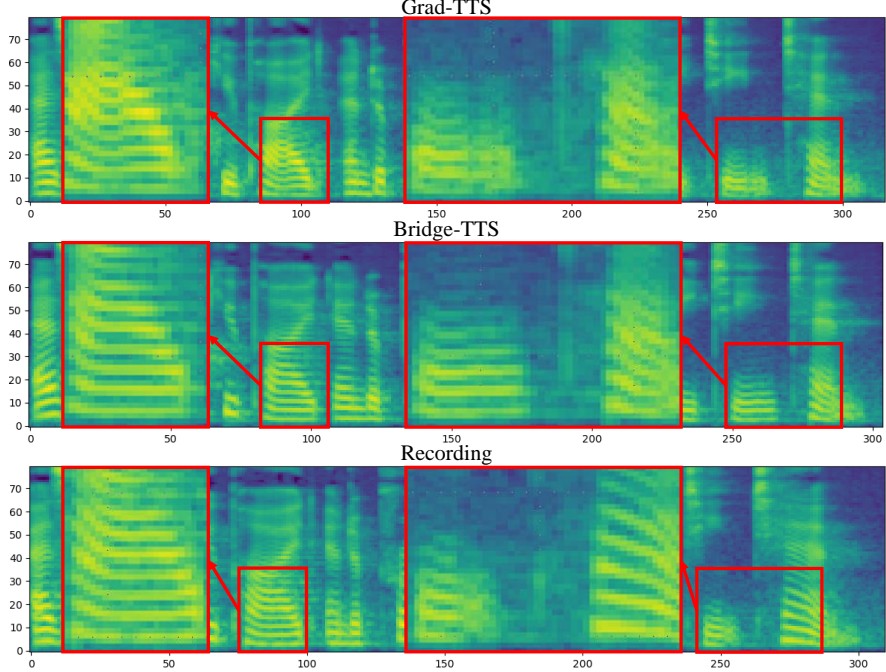

Figure 6: The mel-spectrogram of synthesized (NFE=1000) and ground-truth sample LJ002-0029.

**50-Step Generation** In Figure 7, our method shows higher generation quality than Grad-TTS (Popov et al., 2021). In Figure 8, we continue to use the test sample LJ002-0029 to demonstrate our performance. As it can be seen, in comparison with NFE=1000 shown in Figure 6, when reducing NFE from 1000 to 50, Grad-TTS generates fewer details and sacrifices the sample quality, while our method still generates high-quality samples, outperforming our diffusion counterpart.

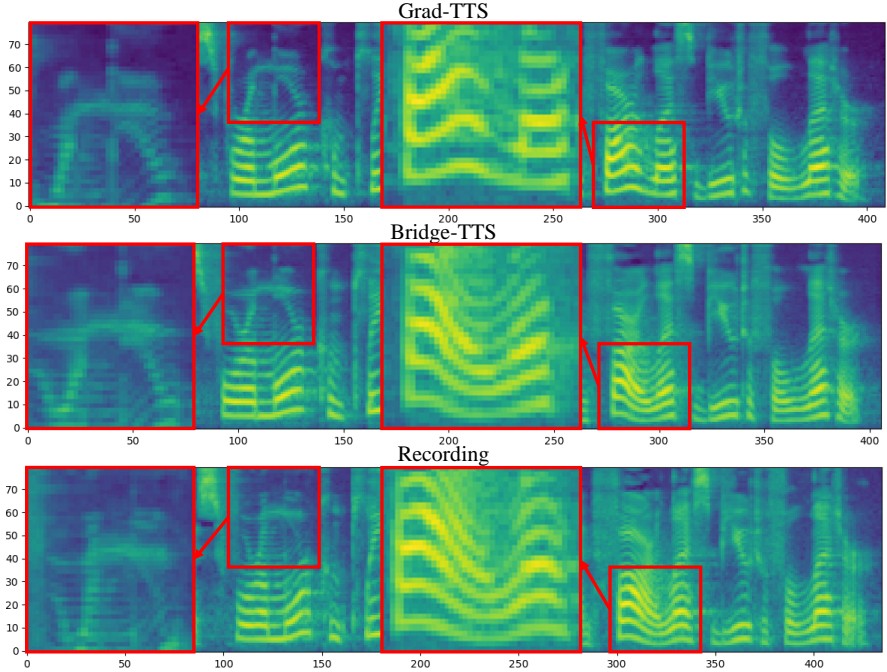

Figure 7: The mel-spectrogram of synthesized (NFE=50) and ground-truth sample LJ001-0035.

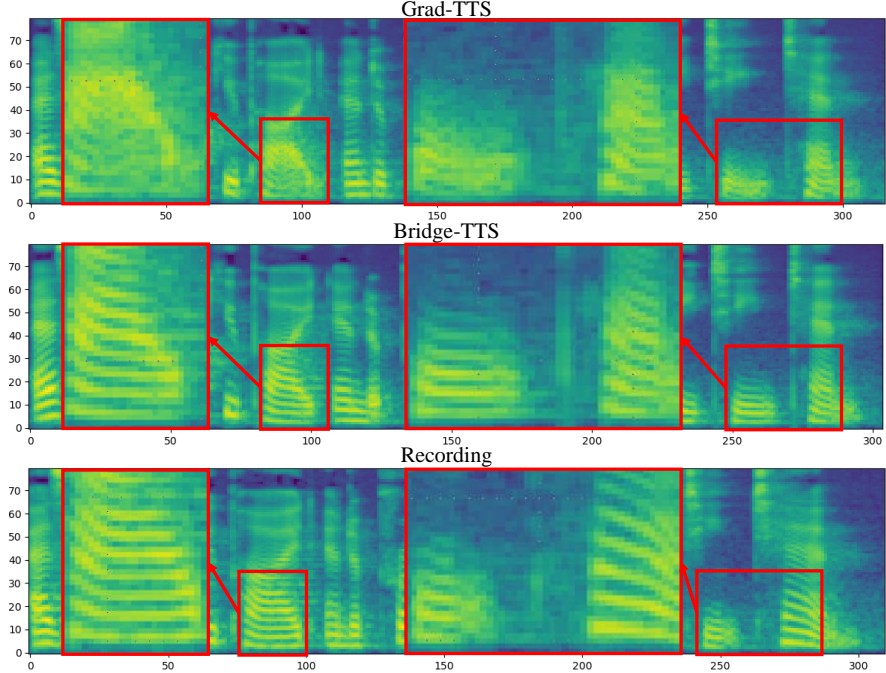

Figure 8: The mel-spectrogram of synthesized (NFE=50) and ground-truth sample LJ002-0029.

**4-Step Generation**   In Figure 9, we show our comparison with two baseline models, *i.e.*, Grad-TTS (Popov et al., 2021) and FastGrad-TTS (Vovk et al., 2022). The latter one employs a first-order maximum-likelihood solver (Popov et al., 2022) for the pre-trained Grad-TTS, and reports stronger quality than Grad-TTS in 4-step synthesis. In our observation, when NFE=4, FastGrad-TTS achieves higher quality than Grad-TTS, while our method Bridge-TTS achieves higher generation quality than both of them, demonstrating the advantage of our proposed data-to-data process on sampling efficiency in TTS synthesis.

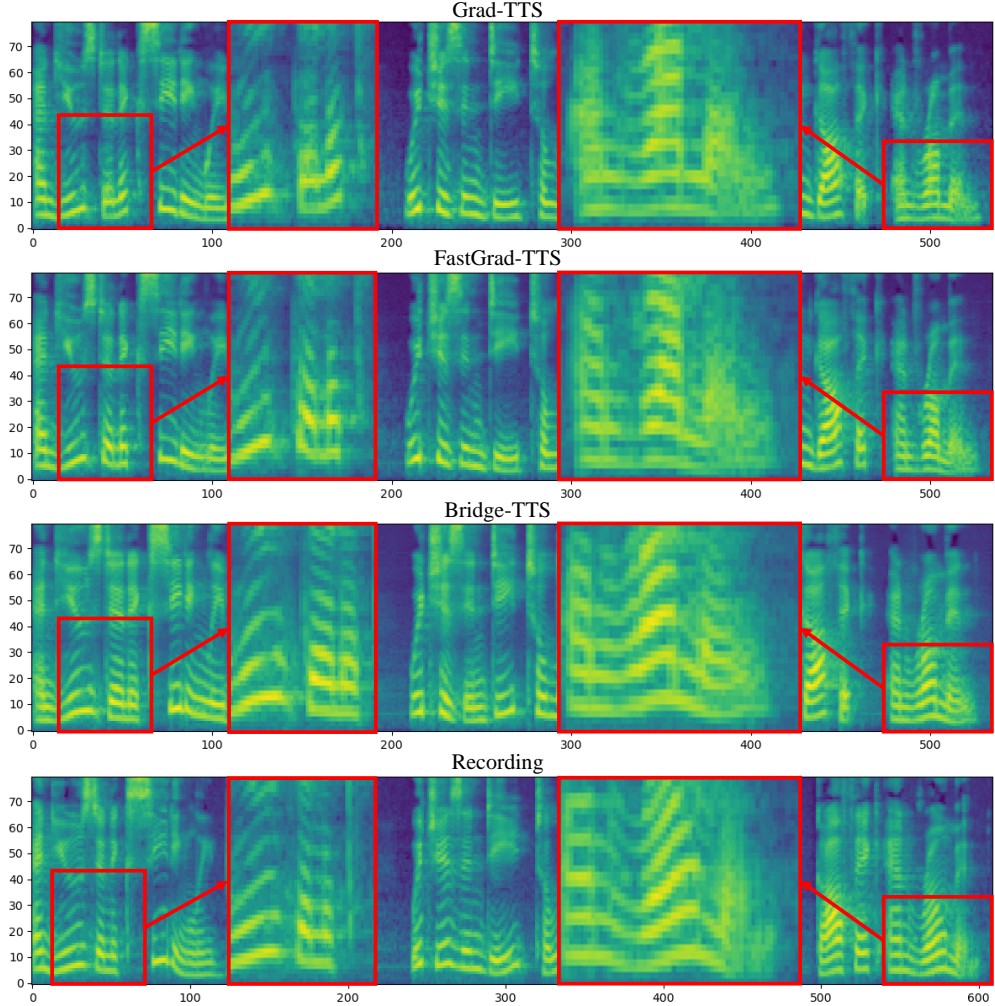

Figure 9: The mel-spectrogram of synthesized (NFE=4) and ground-truth sample LJ001-0032.

**2-Step Generation** When NFE is reduced to 2, we compare our method Bridge-TTS with the transformer-based model FastSpeech 2 (Ren et al., 2021) and two diffusion-based TTS systems using distillation techniques. ProDiff (Huang et al., 2022) employs progressive distillation achieving 2-step generation. CoMoSpeech (Ye et al., 2023) employs consistency distillation achieving 1-step generation. In our observation, in this case, the RTF of each model has been very small, and the overall generation quality is reduced. In the subjective test, our Bridge-TTS outperforms the other three methods. We show a short test sample, LJ001-0002, in Figure 10.

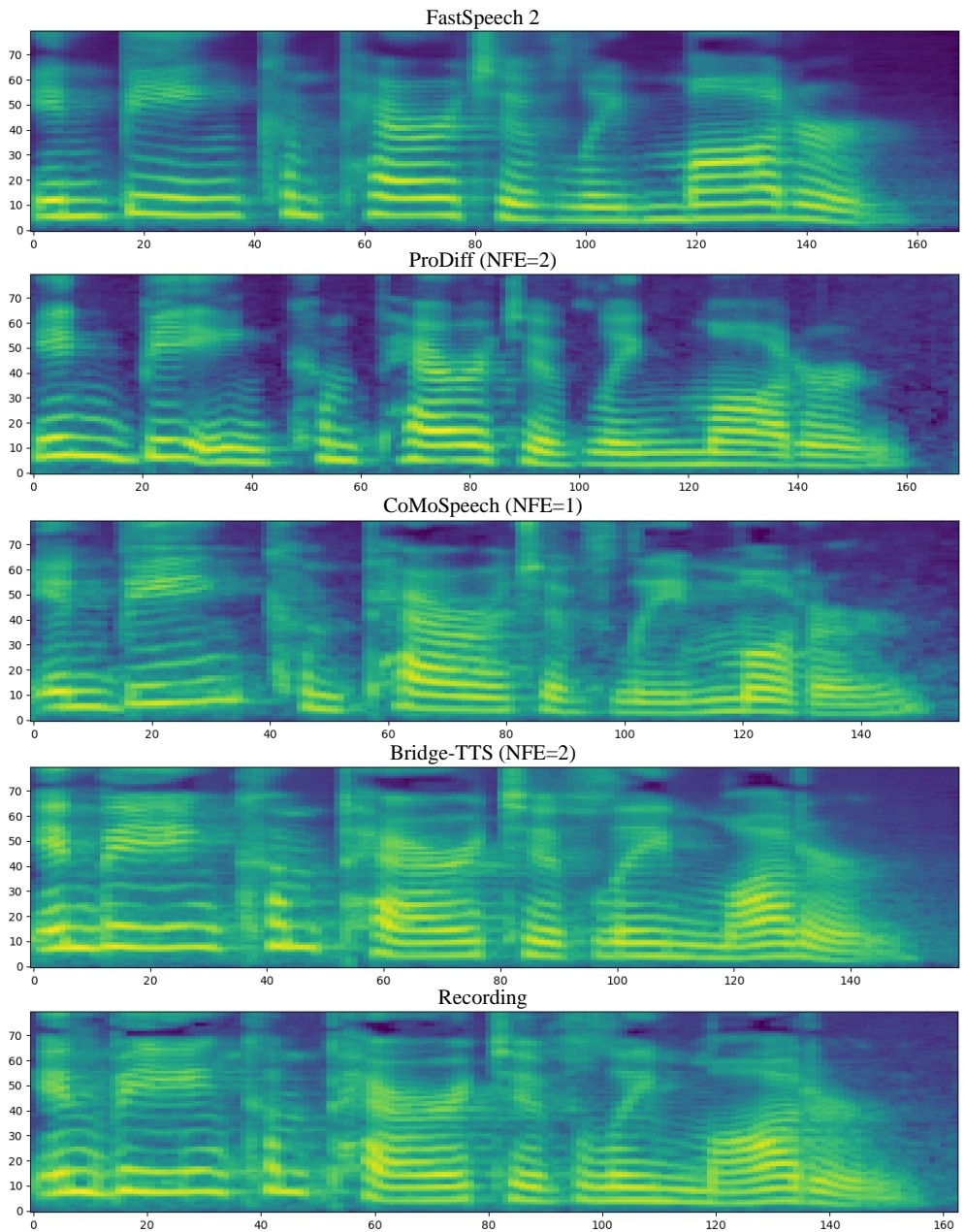

Figure 10: The mel-spectrogram of synthesized (NFE$\leq$2) and ground-truth sample LJ001-0002.

