# OpenReview forum: "Bridge-TTS: Text-to-Speech Synthesis with Schrodinger Bridge"
_ICLR.cc/2024/Conference — ICLR 2024 Conference Withdrawn Submission_

### Official Review · Reviewer_zyj4 · 2023-11-01

**Soundness:** 3 good
**Presentation:** 2 fair
**Contribution:** 2 fair
**Rating:** 5
**Confidence:** 5

**Summary:**

This paper presents Bridge-TTS, which incorporates the Schrödinger Bridge concept into text-to-melspectrogram generation. By introducing the Schrödinger bridge that directly connects the deterministic latent representation from the text encoder and the data, it allows for more direct use of the text encoder output compared to Grad-TTS. It demonstrated better sample quality with fewer sampling steps than Grad-TTS on the LJSpeech dataset.

**Strengths:**

1. This paper applies a strong theoretical background on the Schrödinger Bridge and the recent sampling methods from diffusion models to TTS.

2. Bridge-TTS shows better sample quality with fewer sampling steps compared to Grad-TTS, and even with very few sampling steps (2-step generation), it produces reasonably good quality samples.

**Weaknesses:**

1. The performance improvement compared to Grad-TTS is marginal. Grad-TTS is a paper published in ICML 2021, and improving upon this baseline for a single speaker dataset (LJSpeech) doesn't seem to be a challenging issue in speech synthesis at present and appears to be a straightforward application. Thus, I believe it would be difficult for it to be published in ICLR 2024. Exploring this new generative model seems to have a fresh aspect, so targeting a speech-related venue might be more appropriate. Personally, I feel that the TTS problem for single speaker datasets in the years 2023-2024 is relatively a toy problem. Generating high-quality samples from LJSpeech doesn't appear to be a challenging issue, and I'm not particularly motivated by applying the Schrödinger bridge to TTS given the experimental results in the paper.

**Questions:**

* The paper explores the scalar values of f and g, and shows the CMOS ablation results in Table 4. If the mel-spectrogram data is normalized to have values between [-1, 1] before training, couldn't we simply use 0 for f and 1 for g? By using f=0 and some scaling value for g, this approach appears to be the application of the Brownian bridge by Tong et al to TTS, as also mentioned in the paper.

* The Schrödinger bridge is used in the image domain for applications like Image-to-image translation. If the Schrödinger bridge offers advantages in TTS not only for sampling speed on a single speaker but also for applications like translation in the speech domain (e.g., speech-to-speech translation, voice conversion, etc.), highlighting such results would have provided stronger motivation for its application. This could have made the research more compelling.

---

> ### Author Response · Authors · 2023-11-15
> **Response to Reviewer zyj4**
>
> We appreciate your valuable comments and feedback on our work. We would like to provide our answer and update as follows:
>
> > W1: The performance improvement compared to Grad-TTS is marginal.
>
> **A**: We admit the performance improvement was marginal in our initial submission. **In our latest submission, we outperform our diffusion counterpart Grad-TTS by a large margin and achieve higher quality than the state-of-the-art fast sampling method CoMoSpeech (ACM Multimedia 2023).** Please refer to *Common Response* and our updated submission for more details. We hope this improvement will be considered in your evaluation.
>
> > W2: Targeting a speech-related venue rather than ICLR might be more appropriate.
>
> **A**: We respectfully disagree. This work is not a simple application of known techniques to a speech-related task, instead, it makes **solid theoretical and technical contributions**. Specifically, our proposed tractable SB framework elucidates flexible options for design space, such as the form of noise schedules and SDE/ODE samplers in SB, and they are actively under research (for example, a paper called "Denoising Diffusion Bridge Models" submitted to ICLR 2024 makes many similar contributions to us). Notably, the first-order Bridge ODE sampler has not been revealed by previous works in SB. Therefore, except for the significant results on the TTS task in our revised paper, we believe our methodology itself is general and has future impacts.
>
> > W3: The TTS problem for single-speaker datasets in the years 2023-2024 is relatively a toy problem.
>
> **A**: Grad-TTS is published in ICML 2021. However, to the best of our knowledge, most following diffusion-based TTS systems (acoustic model instead of end-to-end TTS system) mainly work on **fast sampling methods**, and **none of the following works has reported a new record of sample quality in comparison with Grad-TTS-1000**. By working on the training and sampling of the Schrodinger bridge, we achieved a remarkable advancement in both the sample quality and the inference speed compared to diffusion-based TTS systems.
>
> > Q1: By using f=0 and some scaling value for g, this approach appears to be the application of the Brownian bridge by Tong et al. to TTS, as also mentioned in the paper.
>
> **A**: In our latest submission, we have updated Bridge-TTS systems with **asymmetric noise schedules (Bridge-VP and Bridge-gmax)**, which are of importance in improving TTS sample quality and have not been proposed by previous works. In comparison with previous works using Brownian bridges, we ablate the importance of noise schedules (please refer to "constant $g(t)$" in Table 4). By using $f=0$ and some scaling value for $g$, the performance is not as good, and this is in agreement with diffusion models, for which researchers tune the noise schedule and find VP a good choice. We directly use the same hyperparameters as VP, and our performance significantly outperforms Grad-TTS (which also uses VP but under a diffusion backbone). Therefore, we prove that our SB framework is superior to the diffusion counterpart on the TTS task. Moreover, the new noise schedules in the SB framework are non-trivial and distinguished from previous works of Brownian bridges.
>
> > Q2: Other applications like translation in the speech domain
>
> **A**: Thank you for your suggestion. We believe our proposed method could find applications in your mentioned fields, and we would like to explore them in future works. In this work, we expect to focus on our contributions to **improving the sample quality and the inference speed** of diffusion-based TTS systems.

---

> ### Author Response · Authors · 2023-11-18
> **Looking forward to further feedback**
>
> Dear Reviewer zyj4,
>
> We thank you very much again for the great efforts on reviewing our manuscript and providing the valuable comments to further improve. We hope you may find our response and updated submission (paper, website, and synthesized samples) satisfactory, and evaluate our contribution on generative models, TTS sample quality, and inference speed compared to diffusion-based TTS systems.
>
> If you want any further discussion, we would be very happy to reply. If our feedback is good enough, we wonder if you would like to increase your rating accordingly.
>
> Best,
> Authors of Bridge-TTS

---

> > ### Comment · Reviewer_zyj4 · 2023-11-18
> >
> > I have read the updated paper and the reviewers' responses, and I appreciate the considerable effort put into them. I would like to leave a few comments regarding the response.
> >
> > * I believe the experimental results presented in the Bridge-TTS paper don't sufficiently motivate the proposed method. As highlighted in the initial review, several previous works, such as Grad-TTS and VITS, have already demonstrated strong performance on the LJSpeech dataset a few years ago. Considering the significant achievements of these works, most of the recent improvements on LJSpeech tend to fall within confidence intervals and do not offer a significant improvement, which somewhat undermines the motivation for the research method. Enhancing sample quality with fewer sampling steps on the LJSpeech dataset is not enough to justify the proposed approach. It is also noteworthy that no papers focusing exclusively on reducing sampling steps in a single speaker dataset have been published at major conferences like NeurIPS 2022, ICML 2023, ICLR 2023, or NeurIPS 2023.
> >
> > * In the broader context of Text-to-Speech (TTS) advancements, contemporary papers have already shown notable improvements not just in sample quality and sampling speed, but also in multi-speaker and prosody modeling. For the Bridge-TTS method, leveraging the Schrödinger Bridge, to establish its distinct advantage, it would be necessary to demonstrate high sample quality and benefits while maintaining the diversity of prosodies with fewer sampling steps, and notably, in a dataset that presents more complex challenges (e.g. LibriTTS, …).
> >
> > * For a more convincing demonstration of the theoretical contributions, a direct comparison with existing works in the image domain, where Schrödinger Bridge-related research is more mature, would be appropriate. Research on DDBM also provides comparisons with more recent studies like I2SB, Rectified Flow, etc.
> >
> > * Upon reviewing the samples on the demo page, it's noticeable that **Bridge-TTS-gmax-2**, despite achieving a MOS of 4.04, exhibits a somewhat robotic quality.

---

> ### Author Response · Authors · 2023-11-18
> **Response to Reviewer zyj4 (Round 2)**
>
> We have read your comments, and we appreciate your active participation in the author-reviewer discussion period. We would like to leave a few responses regarding the comments for further clarification.
>
> > Comment 1: Enhancing sample quality with fewer sampling steps on the LJSpeech dataset is not enough to justify the proposed approach.
>
> According to Table 1, we have compared the best generation quality of different methods, and in comparison with Grad-TTS-1000 and VITS which achieve strong performance in your comment, we have shown **distinctively better** results instead of falling within confidence intervals (Ground-truth > Bridge-TTS >> other methods). Therefore, we argue that our main contribution is **not only improving sampling speed, but also generation quality**.
>
> > Comment 2: No papers focusing exclusively on reducing sampling steps in a single speaker dataset have been published at major conferences like NeurIPS 2022, ICML 2023, ICLR 2023, or NeurIPS 2023.
>
> As we have claimed, we do not focus exclusively on reducing sampling steps. Besides, other works are not published just because they do not have enough theoretical contributions or do not make significant experimental advancements.
> **It is not reasonable to evaluate a work based on previous works of similar tasks, but not on the quality of the work itself.**
>
> > Comment 3: For the Bridge-TTS method, leveraging the Schrödinger Bridge, to establish its distinct advantage, it would be necessary to demonstrate high sample quality and benefits while maintaining the diversity of prosodies with fewer sampling steps, and notably, in a dataset that presents more complex challenges (e.g. LibriTTS, …).
>
> We highlight the advantages of our data-to-data process over the data-to-noise process in diffusion models. **To the best of our knowledge, LJ-Speech is the only TTS dataset reported in most published diffusion-based acoustic models**, including Grad-TTS (ICML 2021), ProDiff (ACM MM 2022), DiffSinger (AAAI 2022), Fast Grad-TTS (INTERSPEECH 2022) and CoMoSpeech (ACM MM 2023). Hence, we believe this dataset provides a fair comparison between generative models. When a multi-speaker dataset is preferred to demonstrate our contribution to sample quality and inference speed, we wonder if there are appropriate diffusion methods for a fair comparison.
>
> > Comment 4: For a more convincing demonstration of the theoretical contributions, a direct comparison with existing works in the image domain, where Schrödinger Bridge-related research is more mature, would be appropriate. Research on DDBM also provides comparisons with more recent studies like I2SB, Rectified Flow, etc.
>
> We didn't claim that our method can be directly applied to other domains, though we hope so. **Our theoretical assumptions such as paired data are _motivated by the TTS task_ directly**, and the theoretical contributions on the TTS task itself are still valid. With our developed techniques, we have demonstrated our framework's effectiveness, and we have already conducted the ablation studies to compare with I2SB, (e.g., asymmetric schedules vs symmetric schedule in I2SB, data prediction vs noise prediction in I2SB). Also, Rectified Flow is a distillation method based on flow matching. We have talked about flow matching in Appendix D, where we have proved it performs badly on the TTS task. Also, we have compared with distillation-based methods such as CoMoSpeech in few-step scenarios.
>
>
> > Comment 5: Upon reviewing the samples on the demo page, it's noticeable that Bridge-TTS-gmax-2, despite achieving a MOS of 4.04, exhibits a somewhat robotic quality.
>
> It should be noted that, for Bridge-TTS-gmax-2, a MOS of 4.04 is achieved in comparison with other fast sampling methods, supporting our contribution to **efficient sampling instead of generation quality**. Moreover, this result is conducted on Amazon Turk with 20 synthesized samples and 25 Master workers, which is already a large number in comparison with other TTS works.

---

> > ### Comment · Reviewer_zyj4 · 2023-11-23
> >
> > I appreciate the authors' efforts. The authors have reiterated that the quality of samples and generation in the LJSpeech dataset has improved. However, in my view, the results from LJSpeech alone appear to be not a significant improvement. The introduction of Schrodinger Bridge into the 4-step or 2-step generation process would have been more convincing if it had clearly outperformed the existing Grad-TTS 50-step or 1000-step generation. As the authors have stated, it's appropriate to judge this paper on its own results. Yet, I feel that the experimental results presented don't adequately motivate or justify the scalability to larger scales or more diverse datasets with the current experimental results. Although the extensive results support the paper and raise its score, I still consider it below the bar for publication in ICLR, leading me to give this paper a score of 5.

---

> > > ### Author Response · Authors · 2023-11-23
> > > **Thanks for your response**
> > >
> > > Dear Reviewer zyj4,
> > >
> > > Thank you for your feedback and the effort you made in evaluating our updated submission and our responses. We appreciate it very much that you acknowledge our contribution and increase the score accordingly. We think the few-step performance can be further improved by distillation methods, which are left for future work and are not the focus of our paper. For now, we already provide a novel framework with strong performance in both numerous-step and few-step scenarios, without using any fancy tuning tricks. We hope our framework can become a new baseline, which is neat enough for future works to improve.
> > >
> > > Best, Authors of Bridge-TTS

---

### Official Review · Reviewer_8tg3 · 2023-11-01

**Soundness:** 2 fair
**Presentation:** 2 fair
**Contribution:** 2 fair
**Rating:** 5
**Confidence:** 2

**Summary:**

The paper presents Bridge-TTS that generates mel-spectrograms from deterministic text latent representations. To achieve this, the authors first introduce a fully tractable Schrodinger bridge for paired data in TTS modeling. Subsequently, they propose a novel first-order discretization of the Bridge SDE/ODE for accelerated sampling. Experimental results emphasize that the proposed approach offers synthesis quality comparable to or surpassing baseline methods, especially in few-step sampling scenarios.

**Strengths:**

* The authors introduce a theoretically novel approach to employ Schrodinger bridge for TTS that produces outputs from deterministic latent representations.
* They present a new sampling scheme optimized for fast sampling.

**Weaknesses:**

* A lack of empirical validation for the superiority of the proposed method.

It is uncertain whether the proposed deterministic latent representations are superior to the noisy Gaussian conditional prior distribution of diffusion-based TTS. Experimental results suggest that for fewer than 50 sampling steps, the proposed method seems to yield better sample quality compared to other models. However, at 1000 steps, the diffusion-based TTS model, namely Grad-TTS, outperforms the proposed methodology.

Accordingly, for a fair comparison concerning its efficient sampling scheme, it would be appropriate to contrast it with diffusion-based TTS models using a sampling scheme like the DDIM.

**Questions:**

It would be essential for the authors to provide an explanation for the observed worse sample quality of their proposed method compared to the baseline diffusion-based approach at 1000 sampling steps.
Additionally, the generation quality between the proposed method and diffusion-based methods using a comparable sampling scheme in few-step sampling scenarios would also be an interesting aspect of this research.

typo: (p.7, Sec.4.1.,) English graphme -> English grapheme

---

> ### Author Response · Authors · 2023-11-15
> **Response to Reviewer 8tg3**
>
> We are grateful for your valuable comments and feedback on our work. We would like to provide our answer and update as follows:
>
> > W1: A lack of empirical validation for the superiority of the proposed method.
>
> **A**: We have significant advancement in our experiment results and achieve distinctively better quality than Grad-TTS at 1000 steps. Please refer to *Common Response* and our updated submission for more details. Our updated synthesized samples can be visited in the supplementary materials and the anonymous website.
>
> > W2: Compare to diffusion-based TTS models with a comparable sampling scheme
>
> **A**: To demonstrate our performance on efficient sampling, we include two new baseline methods: Fast Grad-TTS (INTERSPEECH 2022), which uses a first-order SDE sampler, and CoMoSpeech (ACM Multimedia 2023), which uses consistency distillation. The first-order SDE sampler is proposed by the authors of Grad-TTS, while we observe the improvement is limited because of the difficulty of generating clean samples from noisy prior. In our updated submission, given the same NFE or similar RTF, we have achieved higher sample quality than different fast sampling methods.

---

> ### Author Response · Authors · 2023-11-18
> **Looking forward to further feedback**
>
> Dear Reviewer 8tg3,
>
> We thank you very much again for the great efforts on reviewing our manuscript and providing the valuable comments to further improve. We hope you may find our response and updated submission (paper, website, and synthesized samples) satisfactory, and evaluate our contribution on generative models, TTS sample quality, and inference speed compared to diffusion-based TTS systems.
>
> If you want any further discussion, we would be very happy to reply. If our feedback is good enough, we wonder if you would like to increase your rating accordingly.
>
> Best,
> Authors of Bridge-TTS

---

> ### Author Response · Authors · 2023-11-20
> **Looking forward to further discussion**
>
> Dear Reviewer 8tg3,
>
> As the remaining time for discussion is limited, we sincerely want to hear your feedback on our updated submission. If you have any further question, we would be very happy to reply.
>
> Thank you very much again for your time, great efforts, and valuable comments.
>
> Best regards, Authors of Bridge-TTS

---

> > ### Comment · Reviewer_8tg3 · 2023-11-22
> > **Response to the authors**
> >
> > It's positive that the authors have provided updated experimental results supporting the proposed model's performance at both high and low sampling steps. This can now be recognized as a part of the empirical contribution of this work, particularly in terms of the efficacy of an efficient sampling scheme. However, the experimental results with numerous sampling steps (Table 2) show that almost all models, including ground truth mel-spectrogram (gt-mel) with vocoder, have overlapping confidence limits or are even reported higher. This scenario makes it difficult to assert that the model significantly outperforms others, as the majority of the scores are very high and almost all models perform on par with or better than gt-mel + vocoder. Given this lack of clear superiority, I am not currently inclined to switch to a favorable review.

---

> > > ### Author Response · Authors · 2023-11-22
> > > **Thanks for your response**
> > >
> > > Dear Reviewer 8tg3:
> > >
> > > We sincerely appreciate your feedback and comments on our revised submission. We would like to make some extra clarifications about our empirical superiority on sample quality with numerous sampling steps:
> > >
> > > First, we would like to clarify that the ground-truth baseline to look at should be the _Recording_, which is consistent with previous TTS works; e.g., Grad-TTS and VITS refer to the recording as ground-truth. The gt-mel + vocoder here is a baseline for reference rather than a primary focus for comparison. Given that, for large sampling steps, the quality of our method is evidently closer to the ground truth recording than other baselines, demonstrating our superior quality. Moreover, compared with GradTTS-1000, a gain of more than 0.07 for BridgeTTS-1000 on MOS is fair to show a discernible quality difference with the crowd-sourced test for TTS.
> > >
> > >
> > > Second, we would like to address that the overlapping confidence interval of MOS between different models does not necessarily suggest that our improvement is marginal due to the nature of crowd-sourced evaluation. In order to supplement the MOS result, we also show the result of a preference test in Appendix G.1, where 11 judges are invited to conduct a blind test between Grad-TTS-1000 (i.e., our diffusion counterpart) and Bridge-TTS-1000 (gmax). As shown in Figure. 4a, **none of them** voted for Grad-TTS-1000.
> > >
> > > Apart from the results of the MOS and preference test, we have provided the synthesized samples (i.e., 14 samples per method) in the supplementary material and presented several demos on our website. We kindly request you visit them and draw a conclusion on the perception quality.
> > >
> > > Again, thank you for your response and active engagement. We look forward to your feedback and are glad to answer any further questions.
> > >
> > > Best regards,
> > >
> > > Authors of Bridge-TTS

---

> ### Author Response · Authors · 2023-11-23
> **Looking forward to post-rebuttal feedback!**
>
> Dear Reviewer 8tg3,
>
> **Your concern on sample quality when NFE=1000 is exactly our major objective during 29-September-2023 to 15-November-2023**. At this final stage of the author-reviewer discussion, we would like to add further evidence showing our sample quality improvement and emphasize our proposed techniques for achieving it.
>
> We conduct a CMOS test on Amazon Mechanical Turk between Grad-TTS-1000 (i.e., our diffusion counterpart) and our Bridge-TTS-1000 (gmax), where 15 Master judges compare 20 samples synthesized by these two methods. As a result, we achieve **a CMOS of +0.21** in comparison with Grad-TTS (**CMOS > +0.1 implies distinctively better**). We wonder if this result and our preference test shown in Appendix G.1 could be evidence for our contribution to sample quality.
>
> Moreover, we expect to emphasize the three techniques for improving the sample quality, which is the main content in our updated submission: 1) we developed asymmetric noise schedules (i.e., gmax and VP) that are strongly by the TTS task itself and are presented for the first time in bridge-related research works; 2) we change the model parameterization from noise prediction to data prediction; 3) we verified the function of sampling temperature in Bridge-based TTS system. Each of these three techniques has been verified in ablation studies with the CMOS test.
>
> We wonder if the above explanations could be helpful to your concerns on sample quality and to clarify our contributions. If you have any questions, we would be very happy to reply. If our feedback is good enough, we wonder if you would like to increase your rating accordingly.
>
> Best regards,
> Authors of Bridge-TTS

---

> > ### Comment · Reviewer_8tg3 · 2023-11-23
> > **Response to the authors**
> >
> > I find the theoretical contribution of this study to be sufficiently interesting, and I am genuinely grateful to the authors for providing new experimental results up to the rebuttal period to support their findings. In particular, when considering CMOS comparison with a strong baseline and the performance of the demo samples, it appears that this method outperforms the strong baseline. However, as I have previously noted, the proposed method, like most comparison models, uses a pretrained vocoder that is trained with ground-truth mel-spectrograms. Despite this, most models show overlapping confidence limits with gt-mel + vocoder, or are even rated higher. Therefore, while I retain my current evaluation, I would deem it a sufficient contribution for a conference paper if the study reasonably explains why methods using vocoders are evaluated to be better than those generated from gt-mel and vocoder, or if it is restructured to showcase the superiority of this methodology over existing diffusion models in an alternative manner.

---

> ### Author Response · Authors · 2023-11-23
> **Thanks for your response**
>
> Dear Reviewer 8tg3:
>
> Our response to your concern about gtmet + vocoder is that, as we adopt the human subjective tests MOS and CMOS to evaluate TTS sample quality, the results are not directly decided by objective metrics like a reconstruction task. We honestly report the results conducted by 25 Master judges on Amazon Mechanical Turk (6000 scores for each Table 2 and Table 3), and do not think gtmel + vocoder should necessarily achieve better results than other synthesized samples.
>
> Best regards,
> Authors of Bridge-TTS

---

### Official Review · Reviewer_SrNQ · 2023-11-05

**Soundness:** 3 good
**Presentation:** 3 good
**Contribution:** 3 good
**Rating:** 5
**Confidence:** 4

**Summary:**

The paper presents a novel neural TTS system based on SB. Informative prior in generative models is an important technical point that is well handled by this paper. In certain scenarios, they show improvement over SOTA methods like Grad-TTS and DiffSinger. The proposed tractable SB is created by defining a reference SDE in alignment with diffusion models. Bridge sampling is discussed in this context to generate the target when trained with paired data (clean text, mel spectrogram). Real-time Factor (RTF) is also discussed alongside MOS, CMOS in evaluation on LJSpeech. Proposal shows improvement in reducing high-frequency artifacts.

**Strengths:**

1. Novelty: Introducing SB to TTS domain.
2. Technical rigor
3. Ablation studies

**Weaknesses:**

1. Generalizability of the proposed method is an issue. We don't know if this works on other test sets, other speakers, etc. I am not confident if this model is vast improvement in the TTS research space.
2. Focus of paper seems to be on technique. TTS-related discussion is lesser than expected.
3. "NFE" is not defined. For new audience, it could be an issue.
4. References are needed to say that some improvement in MOS is actually significant (which is what authors are conveying).
5. Some more intuition on SB would be nice for audience with less background knowledge. Maybe a graph of training with loss or some term going down.

**Questions:**

1. I would like to know if pre-trained models (text encoders) can be leveraged to fasten or improve the proposed SB solution.
2. Would authors like to comment on phoneme-level improvement or provide some information on trends?
3. Any word on if there is a trade-off between the quality of the synthesized speech and the computational efficiency of the method?
4. Can you add some additional consistency loss term which can further brings artifacts down?

---

> ### Author Response · Authors · 2023-11-15
> **Response to Reviewer SrNQ**
>
> Thank you for your valuable comments and feedback on our work. We would like to provide our answer and update as follows:
>
> > W1: I am not confident that this model is a vast improvement in the TTS research space.
>
> **A**: In our previous submission, our improvement compared to diffusion models was limited. In our updated version, we have proposed new techniques, added more baseline methods, and conducted a comprehensive evaluation with 25 Master judges on Amazon Turk, achieving significantly better quality than diffusion-based TTS systems. Please refer to *Common Response* and our updated submission for more details.
>
> > W2: The focus of the paper seems to be on technique. TTS-related discussion is less than expected.
>
> **A**: We have revised our related work section and tried to include more TTS-related discussion. If it is expected, we are willing to add more TTS discussion in the appendix. We were hoping to elaborate on our framework and the formulation of the data-to-data process in the main content.
>
> > W3: "NFE" is not defined. For new audiences, it could be an issue.
>
> **A**: Thank you for your considerate advice! We have added the definition of NFE in Sec 4.2.
>
> > W4: References are needed to say that some improvement in MOS is significant (which is what the authors are conveying).
>
> **A**: MOS and CMOS tests are the main evaluation metrics in TTS synthesis, which could be seen in FastSpeech 2 (ICLR 2021), Grad-TTS (ICML 2021), VITS (ICML 2021), DiffSinger (AAAI 2022), NaturalSpeech, and CoMoSpeech (ACM MM 2023). In our updated version, to guarantee the reliability of evaluation results, we conduct the MOS and CMOS tests on Amazon Mechanical Turk with 25 and 15 Master workers, respectively, and demonstrate significantly better results than diffusion-based systems.
>
> > W5: Some more intuition on SB would be nice for the audience with less background knowledge. Maybe a graph of training with loss or some term going down.
>
> **A**: Thanks for your suggestion. We modify Figure 1 to give a more intuitive overview of the SB framework compared to diffusion models. We also add an overview of our methods on the anonymous website. There are also some intuitive comparisons between SGMs and SB in the last paragraph of Sec 2.3. Our training loss is currently a simple data prediction MSE (Eqn. (13)), and we are sorry that we have no room to plot the training curves in the main text, given the limitation of 9 pages.
>
> > Q1: I would like to know if pre-trained models (text encoders) can be leveraged to fasten or improve the proposed SB solution.
>
> **A**: Using pre-trained text encoders can improve the sample quality in our TTS system. We provide the ablation study in our updated version, i.e., mutable prior and fixed prior.
>
> > Q2: Would authors like to comment on phoneme-level improvement or provide some information on trends?
>
> **A**: Phoneme-level improvement is not specially designed in our framework. We use the same phoneme encoder and duration predictor in Grad-TTS. However, the sample quality of Grad-TTS may be restricted by its prior noise distribution, as it only contains a denoising generation process. As our SDE sampling process includes an encoding (i.e., adding noise) stage and a decoding (i.e., denoising) stage, our sample may suffer less restriction of the prior (i.e., the text encoder output).
>
> > Q3: Any word on if there is a trade-off between the quality of the synthesized speech and the computational efficiency of the method?
>
> **A**: As shown in our updated Table 2, there is a trade-off between the sample quality and computational efficiency of our method. The sample quality degrades as NFE becomes smaller. However, our method is more effective than diffusion models in maintaining the quality when reducing the NFE.
>
> > Q4: Can you add some additional consistency loss terms that can further bring artifacts down?
>
> **A**: Thank you very much for your kind suggestion. For a fair comparison with the data-to-noise process in diffusion models, we keep the single regression loss in our updated version, demonstrating the advantage of the data-to-data process in the Schrodinger bridge. In the future, we would like to study adding additional consistency loss or other loss terms to further enhance the performance of our system.

---

> ### Author Response · Authors · 2023-11-18
> **Looking forward to further feedback**
>
> Dear Reviewer SrNQ,
>
> We thank you very much again for your positive feedback, the great efforts on reviewing our manuscript and providing the valuable comments to further improve. We hope you may find our response and updated submission (paper, website, and synthesized samples) satisfactory, and evaluate our contribution on generative models, TTS sample quality, and inference speed compared to diffusion-based TTS systems.
>
> If you want any further discussion, we would be very happy to reply. If our feedback is good enough, we wonder if you would like to increase your rating accordingly.
>
> Best,
> Authors of Bridge-TTS

---

> ### Author Response · Authors · 2023-11-20
> **Looking forward to further discussion**
>
> Dear Reviewer SrNQ,
>
> As the remaining time for discussion is limited, we sincerely want to hear your feedback on our updated submission. If you have any further question, we would be very happy to reply.
>
> Thank you very much again for your time, great efforts, and valuable comments.
>
> Best regards, Authors of Bridge-TTS

---

> ### Author Response · Authors · 2023-11-23
> **Looking forward to post-rebuttal feedback!**
>
> Dear Reviewer SrNQ,
>
> At the final stage of the author-reviewer discussion, we would like to kindly ask if our replies have addressed your concerns in the first-round review.
>
> On 15-Nov-2023, we updated our submission with overall improvement on the experiment results, and our proposed Bridge-TTS achieved significant progress compared to the diffusion counterpart Grad-TTS in both numerous-step and few-step scenarios, which is demonstrated by the improved MOS score and further validated by the preference test in Appendix G.1.
>
> At last, we would like to emphasize our theoretical contributions. Apart from the bridge sampling, the formulation of our framework is novel itself, which supports the design of asymmetric noise schedules (i.e., gmax and VP) and are presented for the first time in bridge-related research works. We believe that our elaborated theoretical framework and the empirical significance distinguish us from those works that directly apply known techniques to a new task. For now, we have already presented a novel framework with strong performance in both numerous-step and few-step scenarios without using any fancy tuning tricks. We hope our framework can become a new baseline that is neat enough for future works to improve.
>
> We would like to know if you have any additional feedback on our revised submission, and we would be happy to address any further questions. If our feedback is good enough, we wonder if you would like to increase your rating accordingly.
>
> Best regards,
> Authors of Bridge-TTS

---

> ### Comment · Reviewer_SrNQ · 2023-11-23
> **Response to the authors**
>
> I appreciate the efforts put by authors in addressing my concerns. Unfortunately, my confidence has decreased on the empirical generality of this method on new domains. Also, I am not able to verify the correctness of all theoretical claims. (Although this doesn't affect my score). In my humble (and perhaps incorrect) opinion, authors may target a more theory-oriented venue for better reception of their method OR demonstrate this method works on multiple domains. Currently, authors are mixing strong theoretical claims (which seems to be domain independent) with highly practical problem of TTS. This is a tricky mix to explore in a paper. Alternatively, I think authors could have chosen image modality for their work as audio domain may need more empirical evaluations to sell the idea (again, this could be a less informed opinion of mine).
>
> Finally, I also don't know if in Table 2 if MOS improvement from 3.98 to 4.05 with no change in (high) standard deviation of 0.07 is enough for ICLR.

---

> ### Author Response · Authors · 2023-11-23
> **Thanks for your response**
>
> Dear Reviewer SrNQ,
>
> We appreciate your feedback on our updated submission. We were hoping to provide our answers as follows:
>
> At first, when submitting our paper to ICLR, we have chosen the generative model track, welcoming your mentioned theory-oriented evaluation.
>
> Second, as we have clarified, we didn't claim that our method can be directly applied to other domains, though we hope so. **Our theoretical assumptions such as paired data are motivated by the TTS task directly**, and the theoretical contributions on the TTS task itself are still valid. We are not "mixing strong theoretical claims with highly practical problem of TTS". About the reason why comparing results on the LJ-Speech dataset, we follow all the published methods that improve diffusion-based acoustic model and adopt their report training and sampling settings, in order to conduct a fair comparison between different methods.
>
> Third, we were hoping to claim that **a CI of 0.07 is not a high number**, which could be verified in other TTS papers. Though other reviewers are concerned about the performance improvement, we have clarified the misunderstandings thoroughly to verify that we do distinctively outperform previous methods.
>
> We are sincerely grateful for your effort in reviewing our paper again, while your rating has actually decreased instead of the confidence.
>
> Best regards,
> Authors of Bridge-TTS

---

### Official Review · Reviewer_vLu2 · 2023-11-07

**Soundness:** 2 fair
**Presentation:** 3 good
**Contribution:** 2 fair
**Rating:** 5
**Confidence:** 3

**Summary:**

This paper presents a non-autoregressive TTS model called Bridge-TTS, which is build on the Schrodinger bridge (SB). Bridge-TTS follows the two-stage TTS pipeline, i.e., the TTS system comprises with a text-to-spectrogram acoustic model and spectrogram-to-wave vocoder model, where the Schrodinger bridge is used in modeling the former. Unlike most diffusion-based TTS models, Bridge-TTS uses deterministic prior, which is learned from the text input in a deterministic way via a text encoder module. Bridge-TTS is able to use diffusion-TTS-like sampling procedure to synthesis samples from the prior, where different SDE/ODE-based samplers can be adopted to trade-off the inference speed and the sample quality. In general, this submission makes a clear presentation, making detailed and well-structured explanations from the theories of diffusion models to that of the Schrodinger bridge, and decent derivation of the methodology in adopting SB in paired data modeling, e.g., TTS task, as long as the training objective and different sampling schemes. A singer-speaker TTS experiment using the well-known TTS benchmark corpus LJ-Speech is conducted to verify the arguments by the paper.

**Strengths:**

- This paper introduces yet another generative model, i.e., the Schrodinger bridge, for tackling the TTS task.
- This paper presents derivations for bridge sampling in the context that the number of sampling steps is small for the first time, and gives exact solution and 1st-order discretization of SB SDE and ODE, allowing for efficient sampling with SB-based generative models. Relationship of the solution to some famous sampling schemes, such as DDIM, is also presented.
- This submission has source codes, which could be helpful for reproducing the results presented.

**Weaknesses:**

Novelty:

The methodology of Bridge-TTS introduced in this submission does not attempt to address the most urgent issues of in the TTS research field, e.g., the prosody modeling, and the paper doesn’t even specify which duration modeling and text-spectrogram alignment scheme are employed. Moreover, the contribution is incremental since this paper introduces yet another kind of generative model into TTS and does not receive significant performance improvement according the experimental results presented. This submission argues that  replacing the noisy prior in previous systems with the clean and deterministic prior can boost the TTS sample quality and inference speed. However, similar arguments have been made and verified in previous works, such as DiffSinger and DiffGAN-TTS. If we look at the training scheme and loss objective carefully, the text encoder output $z$ is in fact the coarse predicted Mel spectrogram as in DiffSinger, which is learned by using the simple MSE-based reconstruction loss. The SB-based module is indeed a spectrogram post-processing module or a “spectrogram super-resolution module”, and can only refine the details of the produced spectrogram and can not fully leverage the generative modeling power of diffusion-based or SB-based models. In this regard,  the contribution of this paper is not sufficient and only incremental.

Experiments:

- Only conducted single-speaker TTS on the LJ-Speech corpus: this is not sufficient since TTS models have reached human-level quality on this data, e.g., NaturalSpeech and StyleTTS-2. It will be more sound if multi-speaker or even multi-emotion TTS experiments are conducted.
- The reason for explaining why Grad-TTS with 1000 NFEs has higher MOS score than that of Bridge-TTS with 1000 NFEs is not convincing.

**Questions:**

- Why the MOS scores of “Recording” and “GT-Mel-Voc.” in Table 2 are so different from those in Table 3?
- Why Grad-TTS (NFE=1000) is faster than Bridge-TTS (NFE=1000) in terms of RTF, and in other cases such as NFE=50 and 4, Grad-TTS is slower than or has equal RTF to Bridge-TTS?
- Why do you think SB-based spectrogram post-processor is better than diffusion-based ones, e.g., coarse predicted Mel spectrogram as condition to Grad-TTS decoder? Is there an intuitive explanation?
- How do you align text and spectrogram during training and how do you model phoneme durations?

Typos and minor edits in presentation:

- There is no definition of $\Psi$ and $\hat\Psi$.
- I think the symbols of “forward score” and “backward score” in the title of Table 1 are reversed.
- “In practice, we prefer the noise prediction” → “In practice, we prefer to the noise prediction”

---

> ### Author Response · Authors · 2023-11-15
> **Response to Reviewer vLu2**
>
> Thank you for your valuable comments and feedback on our work. We would like to provide our answer and update as follows:
>
> > W1: The contribution is incremental since this paper introduces yet another kind of generative model into TTS and does not receive significant performance improvement according to the experimental results presented.
>
> **A**: We admit our contribution was incremental in the previous version. Recently, we have made great progress on both sample quality and inference speed, demonstrating stronger quality than Grad-TTS and the mentioned DiffSinger and DiffGAN-TTS. Please refer to *Common Response* and our updated submission for more details. Moreover, in comparison with them, we are proposing a TTS system built on a data-to-data process, which is different from simply making the noisy prior distribution more informative. Also, we do not require training an additional acoustic model and finding the intersection point like shallow diffusion systems.
>
> > W2: The SB-based module is indeed a spectrogram post-processing module or a “spectrogram super-resolution module” and can only refine the details of the produced spectrogram and can not fully leverage the generative modeling power of diffusion-based or SB-based models.
>
> **A**: At first, as our text encoder is adopted from Grad-TTS which includes a phoneme encoder and duration predictor, its output cannot provide reasonable quality like a pre-trained acoustic model. Moreover, as our SDE generation process includes both an encoding (adding noise) stage and a decoding (denoising) process, our method is not always a post-processing or super-resolution module. We have shown the advantages of our data-to-data process on both generation quality and sampling speed in our updated version and we kindly request you to re-evaluate our method based on the new experimental results.
>
> > W3: Only conducted single-speaker TTS on the LJ-Speech corpus.
>
> **A**: NaturalSpeech and StyleTTS-2 are end-to-end TTS systems requiring complex loss functions, longer training time, and more computational resources in training. Meanwhile, our focus is on demonstrating the advantages of the proposed data-to-data generation process over the data-to-noise process in diffusion models. In most diffusion-based TTS systems like Grad-TTS (ICML 2021), DiffSinger (AAAI 2022), ProDiff (ACM MM 2022), CoMoSpeech (ACM MM 2023), and your mentioned NaturalSpeech, LJ-Speech is the only reported TTS dataset, providing a fair comparison. We have shown significantly better quality than other diffusion-based acoustic models in our updated version. In the future, we will study the end-to-end TTS system with our method and find multi-speaker TTS baseline methods to demonstrate our performance.
>
> > W4: The reason for explaining why Grad-TTS with 1000 NFEs has a higher MOS score than that of Bridge-TTS with 1000 NFEs is not convincing.
>
> **A**: We agree with your comment. We have proposed new technologies and updated our results, which are conducted on Amazon Mechanical Turk. We kindly recommend you check our updated results.
>
> > Q1: Why the MOS scores of “Recording” and “GT-Mel-Voc.” in Table 2 are so different from those in Table 3?
>
> **A**: In the previous submission, the invited judges may not be familiar with TTS evaluation, and the results in two Tables are separately evaluated. In our updated version, we have employed 25 Master workers on Amazon Turk to evaluate 20 synthesized samples in order to guarantee the reliability of results.
>
> > Q2: Why Grad-TTS (NFE=1000) is faster than Bridge-TTS (NFE=1000) in terms of RTF, and in other cases such as NFE=50 and 4, Grad-TTS is slower than or has equal RTF to Bridge-TTS?
>
> **A**: We use the first-order sampler, while Grad-TTS uses the Euler sampler. When NFE is 1000, our speed is slightly slower than Grad-TTS. When NFE is 50 or 4, the difference of RTF is not distinctive, and we do not report that Bridge-TTS is faster than Grad-TTS, given the same NFE.
>
> > Q3: Why do you think SB-based spectrogram post-processor is better than diffusion-based ones, e.g., coarse predicted Mel spectrogram as a condition to Grad-TTS decoder? Is there an intuitive explanation?
>
> **A**: This is a great question. Grad-TTS decoder is built on a data-to-noise process, while we are proposing a data-to-data process. Our SDE sampling process adds noise on the prior at first and removes the noise to generate samples. In this way, our generation may be less restricted by the prior (i.e., the coarse predicted mel-spectrogram) than Grad-TTS, as their sampling only contains a denoising process.
>
> > Q4: How do you align text and spectrogram during training, and how do you model phoneme durations?
>
> **A**: For a fair comparison with diffusion models, we use the same phoneme encoder and duration predictor in Grad-TTS, which is leveraged from Glow-TTS.

---

> > ### Author Response · Authors · 2023-11-20
> > **Looking forward to further discussion**
> >
> > Dear Reviewer vLu2,
> >
> > As the remaining time for discussion is limited, we sincerely want to hear your feedback on our updated submission. If you have any further question, we would be very happy to reply.
> >
> > Thank you very much again for your time, great efforts, and valuable comments.
> >
> > Best regards,
> > Authors of Bridge-TTS

---

> > > ### Comment · Reviewer_vLu2 · 2023-11-22
> > > **Response to authors**
> > >
> > > Thank you for your diligent efforts in addressing the concerns raised during the review process. I appreciate the substantial work you have put into improving your paper in response to the feedback. After careful re-evaluation of your revised manuscript and considering the changes made, I acknowledge the progress and improvements you have achieved. The improvements are indeed noteworthy, but they have not sufficiently addressed the critical aspects that influenced my initial evaluation, as mentioned in the initial review comments. In light of these improvements, I have decided to revise my score to a 5.

---

> > > > ### Author Response · Authors · 2023-11-22
> > > > **Thanks for your response**
> > > >
> > > > Dear reviewer vLu2:
> > > >
> > > > We are grateful for your feedback and the effort you made in evaluating our updated submission. We also appreciate your positive comments on our revision. Still, we'd like to make some further clarifications regarding our contributions:
> > > >
> > > > Firstly, we understand that you are concerned about the single-speaker setting. In this work, **as we were hoping to highlight the advantages of the data-to-data process over the data-to-noise process in diffusion models, we follow all of the published baseline methods listed in our work and rebuttal, conducting a fair comparison on the LJ-Speech dataset**. Concurrently, two flow matching-based TTS works (Matcha-TTS and ReFlow-TTS) also compare their methods with diffusion models on this dataset. In the future, we would like to conduct experiments on multi-speaker and multi-emotion TTS datasets.
> > > >
> > > > Secondly, we still would like to stress that our text encoder is only composed of a phoneme encoder and duration predictor, which is not a pre-trained acoustic model. Moreover, the SDE sampling process includes a noise additive process and a denoising process. Based on these two facts, our method is a TTS system based on generative models instead of a post-processing/super-resolution module.
> > > >
> > > > Thirdly, we would like to emphasize our **theoretical contributions**. Our theoretical novelty comes from not only the bridge sampling, as you mentioned, but also **the entire framework and asymmetric noise schedules** (i.e., gmax and VP), which are motivated strongly by the TTS task itself and are presented for the first time. We believe that our elaborated theoretical framework and the empirical significance distinguish us from those works that directly apply known techniques to a new task.
> > > >
> > > > Best regards,
> > > >
> > > > Authors of Bridge-TTS

---

> ### Author Response · Authors · 2023-11-18
> **Looking forward to further feedback**
>
> Dear Reviewer vLu2,
>
> We thank you very much again for the great efforts on reviewing our manuscript and providing the valuable comments to further improve. We hope you may find our response and updated submission (paper, website, and synthesized samples) satisfactory, and evaluate our contribution on generative models, TTS sample quality, and inference speed compared to diffusion-based TTS systems.
>
> If you want any further discussion, we would be very happy to reply. If our feedback is good enough, we wonder if you would like to increase your rating accordingly.
>
> Best,
> Authors of Bridge-TTS

---

### Author Response · Authors · 2023-11-15
**Common Response**

We sincerely thank all reviewers and ACs for your effort and detailed and insightful suggestions. We have important and substantial updates to our paper, and we kindly request you examine these results and consider them when evaluating our work.

## Important Update

Dear reviewers, we would like to respectfully draw your attention to our new experiment results. With our new noise schedules and temperature sampling technique, **We achieve subjective quality close to the ground truth, *significantly* outperforming diffusion-based and other methods**. We believe these results have sufficiently validated our theoretical framework's effectiveness on the TTS task, which you are concerned mostly about and is the main weakness of our previous submission.

We built an **[anonymous website](https://bridge-tts.github.io/)**, and we kindly recommend you check it to get a convenient overview of the current power of our methods.

We also updated **the samples in the supplementary material** for a more thorough comparison between our method and the baselines.

## Summary of Paper Revisions

We highlight the important revisions with the **red** color in the revised manuscript and summarize them below.

**Technical Aspects**:

We made three technical changes based on our established SB frameworks to improve the performance further:
1. The symmetric noise schedule (i.e., $f=0$ with constant $g$) is changed to two asymmetric ones: Bridge-gmax and Bridge-VP;
2. A sampling temperature is used in our SDE sampling process, reducing the variance of added Gaussian noise;
3. Given the above two techniques, the model parameterization is changed from noise prediction to data prediction.

**Experiments**:

We made two changes to the evaluation:
1. The MOS and CMOS tests are conducted by Master workers on Amazon Mechanical Turk now. In the MOS test, 25 Master judges evaluated 20 samples from the 12 methods in Table 1 and Table 2, respectively. In the CMOS test, 15 Master judges evaluate 20 samples synthesized by two different methods.
2.  We add one baseline method, ResGrad, for generation quality comparison and two baseline methods (introduced in Appendix F), FastGrad-TTS (INTERSPEECH 2022) and CoMoSpeech (ACM MM 2023), for fast sampling comparison. Our new technical improvements have outperformed diffusion-based baseline methods by a large margin.

**Content**:

We made several changes to improve the presentation of our manuscript:
1. Remove Table 1 in the previous version, which is somewhat confusing, and update Figure 1 for a more intuitive overview.
2. Introduce duration predictor in the background (Sec 2.2) and training (Sec 3.2).
3. Highlight the design of noise schedules in Sec 3.2.
4. Change the main results (Sec 4.2) and ablation study (Sec 4.3) according to our improvement. And move the ablation and discussion about model parameterization to Appendix D.
5. Update the visualization of noise schedules in Appendix E.
6. Add the result of a preference test in Appendix G.1.
7. Add more visualizations and comparisons of mel-spectrogram in Appendix G.2.


We also address the weaknesses and questions of the reviewers in detail and would like to clarify any further concerns.

---

### Author Response · Authors · 2023-11-23
**Final Rebuttal (1)**

Dear Reviewers, Area Chairs, Senior Area Chairs, and Program Chairs:

We sincerely appreciate the feedback and comment from each reviewer. At this last time of author-reviewer discussion stage, we would like to make the final clarifications regarding our contributions in this work and address the major concerns of reviewers.

### Dataset
We understand the concerns of Reviewer vLu2 and zyj4 that our method does not demonstrate significant improvement on multi-speaker or large-scale dataset. However, we expect to highlight it again: in this work, our main target is to demonstrate the advantages of the **data-to-data process achieved by Schrodinger bridge over the data-to-noise process in diffusion models**. To the best of our knowledge, most published methods improving the diffusion-based TTS system report their performance on this LJ-Speech dataset, including Grad-TTS (ICML 2021), ProDiff (ACM MM 2022), DiffSinger (AAAI 2022), Fast Grad-TTS (INTERSPEECH 2022), CoMoSpeech (ACM MM 2023) and two of our concurrent works Matcha-TTS and ReFlow-TTS. We are also concerned about the quality of multi-speaker TTS system. However, this may not be seen as a the lack of experiment results in Bridge-TTS, as none of previous diffusion-based acoustic models reports a baseline or compares the results in multi-speaker scenarios (e.g., Libri-TTS and VCTK).

### Sample Quality
We totally agree that the sample quality when NFE=1000 should be a major concern of Bridge-TTS. This is exactly the reason supporting us continually to improve our method after the first submission on 29-September-2023. In our update version, we develop three improving techniques on sample quality, making us outperform Grad-TTS-1000 and VITS. To provide a solid result, we not only conduct the MOS test with a large number of Master assessors on Amazon Mechanical Turk, but also conduct a preference test in Appendix G.1 and make a CMOS test between Grad-TTS-1000 and Bridge-TTS-1000 in our most recent reply to Reviewer 8tg3. In all these tests, we achieve better results than our diffusion counterpart, which **we believe are enough to verify that we distinctively outperform other methods**. We also provide additional samples in the supplement and the website. **If the standard metrics, user study and the samples all together cannot prove our performance, what should we believe?**

---

### Author Response · Authors · 2023-11-23
**Final Rebuttal (2)**

### Theoretical Contributions
We admit that the performance on TTS synthesis is the major target of our work. However, we still want to emphasize our theoretical contributions that are especially developed for TTS synthesis. In our updated version, our proposed asymmetric noise schedule gmax has not seen in other tasks, while it is of importance on TTS synthesis quality and sampling speed. Moreover, we present both the first-order and higher-order SDE and ODE Bridge sampler in this work, which is also the first time in bridge-related works.

Besides, as we have clarified, we didn't claim that our method can be directly applied to other domains, though we hope so. **Our theoretical assumptions such as paired data are motivated by the TTS task directly**, and the theoretical contributions on the TTS task itself are still valid.

We believe that our elaborated theoretical framework and the empirical significance **distinguish us from those works that directly apply known techniques to a new task**. For now, we present a novel framework with strong performance in both numerous-step and few-step scenarios without using any fancy tuning tricks. We hope our framework can become a new baseline that is neat enough for future works to improve, e.g., developing bridge distillation methods.

During author-reviewer discussion stage, these are the final clarifications that we could make for our work, Bridge-TTS. We thank the effort from all Reviewers, ACs, SACs, and PCs, and expect our contributions on Schrodinger bridge methods and diffusion-based TTS systems could be recognized.

Sincerely yours,
Authors of Bridge-TTS